# Two-Parameter Flows for Learning Population Dynamics of Physical Systems

**Paul Schwerdtner** [1]    **Tobias Blickhan** [1]    **Benjamin Peherstorfer** [1]

## Abstract

This work addresses the problem of learning the dynamics of high-dimensional probability densities over time using unlabeled samples, without assuming access to trajectory information. We introduce two-parameter flows that learn only sampling-time transports from a base distribution to each marginal and then extract a physics-time velocity by regressing on coupled synthetic trajectories. We prove that the resulting physics-time dynamics are unique and inherit regularity from the sampling-time transports. Because we can build on standard, well-developed conditional flow matching techniques for learning the base-to-marginal transports, our approach scales to high dimensions and avoids per-step optimal-transport couplings, while allowing admissible non-gradient dynamics that can naturally explain rotational or circulating physics phenomena.

## 1. Introduction

### 1.1. Learning Population Dynamics

**Motivation: Surrogate modeling**   Learning dynamics from data is a central challenge in science and engineering, particularly for building surrogate models of complex physical systems. Such surrogates are essential for outer-loop applications such as inverse problems, uncertainty quantification, and design, which require repeated evaluations of the underlying dynamics across many parameter configurations and initial conditions (Peherstorfer et al., 2018). In these settings, relying solely on high-fidelity numerical solvers is often computationally prohibitive. Thus, even if a high-fidelity numerical model is available to generate training data, it is insufficient for solving outer-loop applications because it is too expensive for the many repeated simulations that are needed in the outer loop. The goal is therefore to infer dynamical laws from data that can be used to make fast and accurate predictions for previously unseen parameters and initial conditions within outer-loop applications.

**Sample versus population dynamics**   In deterministic settings where full trajectory data are available, learning the dynamics is often posed as fitting a model by minimizing point-wise discrepancy (e.g., mean-squared residual) along the training trajectories. We refer to this as learning the sample dynamics (or point-wise fitting) because the objective is to approximate the individual training trajectories (Li et al., 2021; Lu et al., 2021; Kovachki et al., 2024). In contrast, we focus on stochastic and chaotic systems, for which learning dynamics of the individual trajectories is often not meaningful and uninformative (see, e.g., Neklyudov et al. (2023); Blickhan et al. (2025) for a discussion). Accordingly, we seek models whose dynamics agree with the training data in law, which means that the goal is not predicting individual sample trajectories but to learn the population dynamics that describe the evolution of the underlying probability law.

**Example: Population dynamics of random walk**   A simple example illustrates both the usefulness of population dynamics and the limitations of sample-trajectory learning. Consider the stochastic differential equation (SDE) $\mathrm{d}\boldsymbol{x}_t = \sigma \mathrm{d}\mathbf{W}_t$ and assume $\boldsymbol{x}_0 \sim \mathcal{N}(0, \boldsymbol{I})$ is independent of the Wiener process $\boldsymbol{W}_t$. The law of the corresponding samples is $\rho(t) = \mathcal{N}(0, (1 + t\sigma^2)\boldsymbol{I})$. The sample trajectories (paths) of this system are almost surely nowhere differentiable, making a point-wise regression on the sample trajectories inherently rough. In contrast, the same marginal evolution can be generated by a deterministic system of ordinary differential equations (ODEs) of the form $\dot{\boldsymbol{x}}_t = u(\boldsymbol{x}_t, t)$ with the velocity $u(\boldsymbol{x}_t, t) = \sigma^2 \boldsymbol{x}_t / (2(1 + \sigma^2 t))$ and initial condition $\boldsymbol{x}_0 \sim \rho(0)$, whose solutions are smooth and given explicitly by $\boldsymbol{x}_t = \sqrt{1 + \sigma^2 t}\boldsymbol{x}_0$. Thus, while the sample trajectories are challenging (and in a precise sense impossible) to capture point-wise, the associated population dynamics admit a simple, smooth representation that is tractable to infer from data.

**Inferring population dynamics**   Given (samples of) time marginals $(\rho(t))_{t \in [0,T]}$ formulated over the domain $\mathcal{X} \subseteq \mathbb{R}^d$ that admit a density $\rho(\cdot, t) : \mathcal{X} \to \mathbb{R}$, we seek a velocity field $u : \mathbb{R}^d \times [0, T] \to \mathbb{R}^d$ such that the induced flow

[1]Courant Institute of Mathematical Sciences, New York University, 251 Mercer Street, New York, NY 10012, USA. Correspondence to: Benjamin Peherstorfer <pehersto@cims.nyu.edu>.

*Proceedings of the 43rd International Conference on Machine Learning*, Seoul, South Korea. PMLR 306, 2026. Copyright 2026 by the author(s).

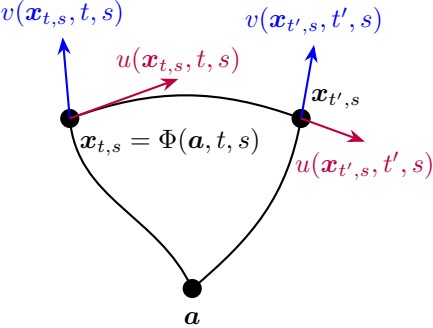

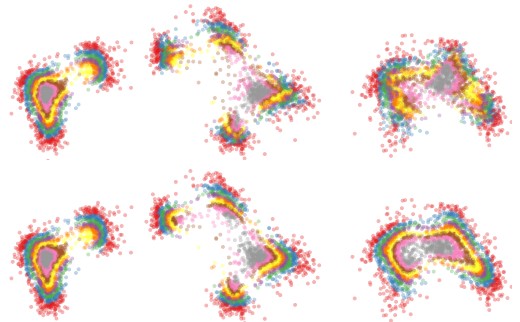

*Figure 1.* Learn vertically, predict horizontally: The vertical flow in sampling time $s$ along $v : \partial_s \Phi = v \circ \Phi$ is learned through existing, scalable methods from generative modeling. After training, we have implicitly defined a horizontal flow $u \circ \Phi = \partial_t \Phi$ which we can make explicit in a second regression step and use for rapid inference in physics time $t$.

*Figure 2.* Snapshots from a curve $t \mapsto \rho(t)$ (left to right: $t \in [0, 1]$). The first row displays piecewise optimal transport: samples retain their color for all $t$ and move between marginals following piecewise OT as in (Terpin et al., 2024). The second row shows that TPF dynamics are similar, but more regular: Especially for later times, the flow defined by successive OT mappings develops sharp gradients in $\mathrm{d}\boldsymbol{x}_{t,1}/\mathrm{d}\boldsymbol{a}$.

ordinary differential equation (ODE)

$$\frac{\mathrm{d}}{\mathrm{d}t}\boldsymbol{x}_t = u(\boldsymbol{x}_t, t), \qquad \boldsymbol{x}_0 \sim \rho(0),$$

induces the prescribed marginals $(\rho(t))_t$ in the sense that $\boldsymbol{x}_t \sim \rho(t)$ for all $t \in [0, T]$. A velocity field $u$ is admissible for $(\rho(t))_t$ if it satisfies the continuity equation

$$\partial_t \rho(\boldsymbol{x}, t) + \nabla \cdot (\rho(\boldsymbol{x}, t) u(\boldsymbol{x}, t)) = 0, \qquad \text{on } \mathcal{X}, \quad (1)$$

in the sense of distributions.

We note that admissible velocity fields are generally not unique. Indeed, if $u$ is admissible and $w$ satisfies $\nabla \cdot (\rho w) = 0$, then $u + w$ is also admissible. This non-uniqueness highlights that learning population dynamics typically requires an additional selection principle (e.g., structural constraints or optimality criteria) to identify a canonical representative among admissible vector fields.

### 1.2. Our Approach: Two-Parameter Flows

We introduce a two-parameter viewpoint to learning population dynamics: we learn the transport of a reference (base) distribution to each time marginal $\rho(t)$, and then use the induced structure to obtain dynamics in the physics-time-direction to evolve $\rho(t)$ over time $t$. The transport from reference to time marginals can be obtained with, e.g., a standard conditional flow matching model and stochastic interpolants (Lipman et al., 2023; Albergo & Vanden-Eijnden, 2023) that dynamically maps $\nu$ to $\rho(t)$ for all $t$.

Our key observation is that once the transport from the base to the $\rho(t)$ for all $t$ is fixed, there is a unique consistent way to connect these transports across physics time $t$. In other words, specifying the flow in one direction (from base to $\rho(t)$) implicitly selects a single, well-defined physics-time velocity among the many admissible ones compatible with the marginals $(\rho(t))_t$ via (1).

We show that the induced physics-time velocity field is guaranteed to produce the correct marginal evolution and it inherits regularity from the transport from base to $\rho(t)$. In particular, we prove regularity bounds for the physics-time velocity field that can be expressed directly in terms of the regularity of the trained (conditional flow matching/stochastic-interpolants) velocity from base to time marginals, which means that if the transport from base to time marginals is regular then so are the dynamics in physics time.

Overall, our two-parameter flow approach provides a principled and scalable pipeline for learning population dynamics: train a sampling-time (vertical) conditional flow with established and mature tools that scale, then extract the corresponding physics-time (horizontal) dynamics for generating new trajectories for new initial conditions and parameters; see Figure 1.

### 1.3. Related Work and Optimal Transport

**Minimal kinetic energy velocity fields** Existing literature focuses to a large part on the minimal kinetic energy field, which is a specific admissible velocity field that is defined as the unique minimizer of

$$\mathcal{E}(u) = \frac{1}{2} \int_0^1 \int |u(x, t)|^2 \rho(x, t) \, \mathrm{d}x \, \mathrm{d}t \qquad (2)$$

while having the continuity equation as a constraint among all admissible $u$. This minimizer is of gradient form so that one can represent it as $u_{\min} = \nabla\varphi$ with a potential $\varphi : \mathbb{R}^d \times [0, T] \to \mathbb{R}$.

**Methods that learn a potential** Neklyudov et al. (2023); Berman et al. (2024); Blickhan et al. (2025) aim to infer this particular minimal-kinetic energy velocity field. In particular, these methods hard-code the gradient structure $u = \nabla\varphi$ by parametrizing and then learning the potential $\varphi$.

**Pairwise optimal-transport couplings**   A different line of work, e.g., (Terpin et al., 2024), is motivated by optimal transport (OT) theory. Here, the OT couplings between the point clouds at successive times is computed. After these couplings have been computed, one can learn a velocity field based on the trajectories defined by successive application of the OT maps. Since infinitesimal optimal-transport maps along a curve $t \mapsto \rho(t)$ converge to the minimal energy admissible vector field in the limit $\delta t \to 0$ (Ambrosio et al., 2005, Proposition 8.4.6), this approach ultimately targets the minimum kinetic energy field that is admissible. Another way to view this particular choice is to learn the potential of a time-dependent Wasserstein gradient flow. These methods are usually named after the Jordan-Kinderlehrer-Otto formalism (Jordan et al., 1998) and have received considerable attention in recent years (Tong et al., 2020; Alvarez-Melis et al., 2022; Bunne et al., 2022; Lavenant et al., 2024; Persiianov et al., 2026).

**Schrödinger Bridges**   The stochastic analogue to JKO methods, Schrödinger bridge matching connects time marginals $\rho(t_j)$ and $\rho(t_{j+1})$ via stochastic processes instead of OT maps (Chen et al., 2019; 2023; Shen et al., 2025; Hong et al., 2025). This approach treats inference as an SDE integration. The resulting vector fields also have gradient form, but follow a distinct selection criterion, see (Léonard, 2014, Definition 2.1) and (Villani, 2009, Chapter 7).

**Autoregressive generative modeling of trajectory data and other related literature**   There is a range of methods (Hoogeboom et al., 2022; Rühling Cachay et al., 2023; Chen et al., 2024; Price et al., 2025) for learning autoregressive diffusion- and flow-based models, building on, e.g., (Sohl-Dickstein et al., 2015; Ho et al., 2020; Hyvärinen, 2005; Song et al., 2019; Song & Ermon, 2019; Song et al., 2021; Albergo et al., 2025; Lipman et al., 2023; Liu et al., 2023). These approaches aim to learn a conditional law that describes the transition from a current state to the next, which is a fundamentally different problem than what we consider and more akin to learning sample-level trajectory dynamics than population dynamics. As a side remark, we note that there are methods that learn the score of a distribution and additionally use knowledge of the drift and/or diffusion coefficient to derive a generative model (Li et al., 2023). We do not have access to the drift and diffusion coefficient in the following, only to data. Furthermore, we note that there is work on learning time-conditioned normalizing flows (Arend Torres et al., 2024); in contrast, our use of transport from the base measure to time marginals is only a scalable first step toward extracting an explicit physics-time velocity.

## 1.4. Two-Parameter Flows and Optimal tTransport

**Why we do not target minimal-kinetic-energy fields**
The optimal-transport/minimal kinetic energy field is mathematically canonical, but it is often not a modeling requirement. As $u$ is not identifiable without additional structure, choosing the minimal-kinetic energy representation is one principled convention, but it is only one convention.

From a learning perspective, targeting the minimal-kinetic energy representation can be unnecessarily restrictive, and even expensive to compute (e.g., computing optimal-transport couplings scales typically cubic in the number of the data points and is intractable for large enough data sets (Persiianov et al., 2026, Section C.2)). Furthermore, adversarial examples exist where the gradient field can be shown to oscillate arbitrarily fast in the case of finite samples (see Gigli (2011), Example 2.4). This shortcoming of existing methods is also pointed out in (Terpin et al., 2024, Section G2). In particular, there is no a priori guarantee that $u$ is smooth (Ambrosio et al., 2005, Section 6.2), see Figure 2.

The recent work (Petrović et al., 2025) provides a mechanism for learning admissible velocity fields with non-gradient components, but it operates in a different regime: it leverages a known reference drift in the form of sample velocities in the training data. In contrast, our setting does not assume access to such a prior. Our approach is fully generic and does not require specifying a reference model for the physics-time velocity field $u$. We note that the works (Benamou et al., 2019; Chewi et al., 2021) replace piecewise optimal-transport couplings with higher-order optimal-transport constructions, such as spline-like curves characterized by minimal acceleration. These formulations can produce temporally smoother population-level trajectories, but they still require solving an OT problem (typically one per marginal or time interval), and thus retain the computational bottleneck associated with OT-based methods.

**Two-parameter flows: admissible and practically learnable but not necessarily of minimal energy**   Our approach does not attempt to recover the optimal-transport/minimal-kinetic-energy velocity field. Instead, we use two-parameter flows to construct an admissible transport from a base distribution to each time marginal and then extract a corresponding velocity field for sampling at inference time. This decouples distribution matching from optimal-transport coupling computation and avoids committing to a gradient, minimal-energy field.

Crucially, many population dynamics are naturally expressed through admissible fields with non-gradient components (e.g., rotational transport), which OT excludes by design. Because our learned field need not coincide with the minimal-kinetic-energy velocity in physics time, it can represent such effects while remaining consistent with the observed marginals. Empirically, this additional flexibility

can also yield smoother fields than OT-based constructions; see again Figure 2.

### 1.5. Summary of Contributions

(a) We introduce two-parameter flows that reduce population dynamics inference to (i) a standard conditional flow training from base to each marginal and (ii) a standard regression step that extracts an explicit physics-time velocity; scaling to turbulence examples in high dimension $> 10^4$.

(b) We prove that the extracted physics-time velocity inherits regularity from the learned sampling-time transport, providing a principled guarantee of well-posed and regular physics-time dynamics.

(c) Unlike optimal-transport/minimal-kinetic-energy approaches, the resulting dynamics are not restricted to gradient flows, which avoids per-step optimal-transport coupling computations and enables broader and often smoother physics-time dynamics.

## 2. Two-Parameter Flows

We introduce two-parameter flows that depend on a sampling time $s$ as well as the physics time $t$. Our aim is to learn only the sampling-direction transport from a base distribution to each time marginal $\rho(t)$. Consistency conditions for regular flow maps then uniquely determine the corresponding physics-time dynamics. We show that the induced dynamics in physics time inherit regularity from the learned sampling-direction transports.

### 2.1. Sampling-Time Velocity and Flow Maps

Recall that we have a time-dependent distribution $(\rho(t))_{t \in [0,T]}$ and a base distribution $\nu$. Furthermore, let us assume we have obtained a velocity field $v : \mathbb{R}^d \times [0,T] \times [0,1] \to \mathbb{R}^d$ so that

$$\frac{\mathrm{d}}{\mathrm{d}s} \boldsymbol{x}_{t,s} = v(\boldsymbol{x}_{t,s}, t, s), \quad \boldsymbol{x}_{t,s}\big|_{s=0} = \boldsymbol{a} \sim \nu, \quad (3)$$

generates samples $\boldsymbol{x}_{t,s}\big|_{s=1} \sim \rho(t)$ at the final sampling time for all $t \in [0,T]$. The velocity field $v$ acts in direction $s$ over $s \in [0,1]$ and is conditioned on $t \in [0,T]$.

We refer to $s$ as sampling time and to $t$ as physics time. Furthermore, to build intuition, it is helpful to visualize the physics time flowing horizontally and the sampling time flowing vertically as in Figure 1.

The velocity field $v$ induces a two-parameter flow $(\boldsymbol{a}, t, s) \mapsto \Phi(\boldsymbol{a}, t, s)$ as

$$\Phi(\boldsymbol{a}, t, s) = \boldsymbol{a} + \int_0^s v(\Phi(\boldsymbol{a}, t, \sigma), t, \sigma) \, \mathrm{d}\sigma, \quad \boldsymbol{a} \sim \nu \ (4)$$

that pushes forward $\nu$ to $\rho(t)$ as $\Phi(\cdot, t, 1)\sharp\nu = \rho(t)$ for all

$t \in [0,T]$. Additionally, the flow satisfies

$$\Phi(\boldsymbol{a}, t, 0) = \boldsymbol{a}, \qquad t \in [0,T], \ \boldsymbol{a} \sim \nu. \quad (5)$$

We call a two-parameter flow $\Phi$ a $C^2$-diffeomorphism if $\Phi$ is twice continuously differentiable in all arguments and $\Phi(\cdot, t, s)$ is a diffeomorphism for all $t \in [0,T]$ and $s \in [0,1]$. Because the regularity of $\Phi$ is governed by the regularity of $v$ (Abraham et al., 1988, Lemma 4.1.9), the flow defined through (4) is a $C^2$-diffeomorphism for sufficiently regular $v$, i.e. of class $C^2$ in all arguments with a global bound on $\nabla v$. We also assume that $\nabla^2 v$ is bounded for convenience.

### 2.2. Physics-Time Velocity $u$

The flow $\Phi$ describes how a sample $\boldsymbol{a} \sim \nu$ moves when we vary either sampling time $s$ or physics time $t$. Correspondingly, we can use $\Phi$ to derive sampling- and physics-time velocity fields: The sampling-time (vertical) velocity is the field $v$ given in (3), which can also be written as

$$\partial_s \Phi(\boldsymbol{a}, t, s) = v(\Phi(\boldsymbol{a}, t, s), t, s). \quad (6)$$

Analogously, we can take the derivative of $\Phi$ in physics time $t$ to define a field $u$ via

$$\partial_t \Phi(\boldsymbol{a}, t, s) = u(\Phi(\boldsymbol{a}, t, s), t, s). \quad (7)$$

The following proposition shows that (7) indeed defines a unique velocity field $u : \mathbb{R}^d \times [0,T] \times [0,1] \to \mathbb{R}^d$ if the flow is sufficiently regular. Furthermore, this velocity $u$ induces a continuity equation for the time marginals $\rho(t)$ (proof in appendix).

**Proposition 2.1.** *Let $v$ be of class $C^2$ and suppose the flow $\Phi$ (4) of $v$ has lifetime one (Abraham et al., 1988, Definition 4.1.15). Then, $\Phi(\cdot, t, s)$ is a $C^2$-diffeomorphism and equation (7) defines a function $u : \mathbb{R}^d \times [0,T] \times [0,1] \to \mathbb{R}^d$, this function $u$ is unique, and it satisfies the continuity equation in a weak sense $\partial_t \rho(t) + \nabla \cdot (\rho(t) u(\cdot, t, 1)) = 0$.*

Notice that (5) with Proposition 2.1 implies $u(\cdot, t, 0) \equiv 0$.

Proposition 2.1 shows that if a sufficiently regular sampling-time velocity $v$ is determined, which transports the base distribution $\nu$ to any marginal $\rho(t)$, then the physics-time dynamics are uniquely determined and the corresponding physics-time velocity $u$ solves (7). In other words, it is sufficient to learn in sampling-time direction $s$ how to transport the base distribution to the time marginals to obtain the physics-time dynamics in form of one particular velocity $u$.

We remark that under the conditions of Proposition 2.1, the flow $\Phi$ also defines a pushforward from $\rho(t)$ to $\rho(t')$ as $\left(\Phi(\cdot, t', 1) \circ \Phi^{-1}(\cdot, t, 1)\right)\sharp\rho(t) = \rho(t')$, which first maps a sample $\boldsymbol{x}_t \sim \rho(t)$ onto a noise sample $\boldsymbol{a}$ of $\nu$ and then transports it back to a sample $\boldsymbol{x}_{t'}$ of $\rho(t')$ at physics time $t'$.

As we will see in Proposition 2.2, the velocity $u(\cdot, t, 1)$ can also be integrated from $t$ to $t' > t$ to transform a sample from $\rho(t)$ into one from $\rho(t')$.

### 2.3. Beyond Conditioning: Flow Consistency in $s$ and $t$

With a two-parameter flow $\Phi$ we can describe how a sample evolves when we vary either the sampling time $s$ or the physics time $t$. In particular, we want that transporting along the vertical $s$-time and horizontal $t$-time directions defines a well-posed two-dimensional motion, rather than two unrelated one-dimensional motions.

We can formalize this notion of consistency between $s$ and $t$ direction via a Schwarz condition as follows: Recall that we have the associated velocities $\partial_s \Phi = v \circ \Phi$ and $\partial_t \Phi = u \circ \Phi$ given in (6) and (7), respectively. A flow $\Phi$ is consistent if the mixed derivatives commute,

$$\partial_t \partial_s \Phi(\cdot, t, s) \equiv \partial_s \partial_t \Phi(\cdot, t, s), \quad (8)$$

for all $t \in [0, T]$ and $s \in [0, 1]$, which formalizes the idea that moving in $s$ and then $t$ equals moving in $t$ and then $s$ when we make infinitesimal steps.

Schwarz' theorem states that mixed derivatives commute if the function is twice continuously differentiable. Thus, if the velocity $v$ is sufficiently regular (see Section 2.1) so that the flow $\Phi$ (as given by (4)) is twice continuously differentiable in $s$ and $t$, then $\Phi$ is consistent in the sense of (8). Furthermore, commuting mixed derivatives (8) lead to the following PDE relating the velocity $v$ and $u$,

$$\partial_s u + v \cdot \nabla u - u \cdot \nabla v - \partial_t v = 0, \quad (9)$$

which is the vanishing Lie bracket condition (Abraham et al., 1988, Section 4.2).

### 2.4. Regularity of Physics-Time Velocity

Using the regularity of $\Phi$ and the concept of consistency, we can derive more properties of the physics-time velocity $u$ defined as $\partial_t \Phi = u \circ \Phi$ besides that it is unique and solves the continuity equation (Proposition 2.1), even if we have learned only $v$ that induces the flow $\Phi$ as with (4). In particular, the consistency relation (9) links derivatives of $u$ to derivatives of $v$, so regularity of $v$ propagates through $\Phi$ and translates into corresponding regularity properties of $u$.

**Regularity of $u$ determined by regularity of $v$** We derive a closed-form expression for $u$.

**Proposition 2.2.** *Let $\Phi$ be a $C^2$-diffeomorphism so that it is consistent and (9) holds. Set $\boldsymbol{x}_{t,s} = \Phi(\boldsymbol{a}, t, s)$. Then, the*

*physics-time velocity defined as*

$$u(\boldsymbol{x}_{t,s}, t, s) = D\Phi(\boldsymbol{a}, t, s)$$
$$\cdot \int_0^s [D\Phi(\boldsymbol{a}, t, \sigma)]^{-1} \partial_t v(\boldsymbol{x}_{t,\sigma}, t, \sigma) \, d\sigma. \quad (10)$$

*satisfies* (7) *and its Jacobian $Du$ is given by*

$$(Du)(\boldsymbol{x}_{t,s}, t, s)D\Phi(\boldsymbol{a}, t, s) = \frac{\mathrm{d}}{\mathrm{d}t} D\Phi(\boldsymbol{a}, t, s).$$

The analytic expression (10) of $u$ directly shows that it inherits regularity from $v$. For example, if $v$ is $k$-times continuously differentiable in the state variable, and at least continuously differentiable in $t$ and $s$ so that $\partial_t v$ is well defined, then $u$ is $k - 1$-times continuously differentiable in the state variable. This loss of one order is due to $D\Phi$ and $(D\Phi)^{-1}$ in (10).

Additionally, the $H^1$-norm of $u$ is bounded as follows:

**Proposition 2.3.** *Let $u$ satisfy the compatibility condition* (9) *with $u_{s=0} \equiv 0$. Then, $\|u\|_{H^1(\rho)}(t, 1) \leq \int_0^1 \|\partial_t v\|_{H^1(\rho)}(t, s) e^{\int_s^1 \left(\frac{1}{2}\|\nabla^2 v\|_\infty + 2\|\nabla v\|_\infty\right)(t,\sigma)\, d\sigma} \, \mathrm{d}s.$*

In practice, we use the flow matching/stochastic interpolant approaches to obtain the field $v$ (Albergo et al., 2025; Lipman et al., 2023; Liu et al., 2023). In this case,

**Proposition 2.4** (Chen et al. (2025), Proposition 2.3)**.** *Let $\nu = \mathcal{N}(0, Id)$ and $\rho(t)$ be given. Define the stochastic interpolant from $\nu$ to $\rho(t)$, with $(\boldsymbol{a}, \boldsymbol{x}_t) \sim \nu \otimes \rho(t)$ and $I_s(\boldsymbol{a}, \boldsymbol{x}_t) = \alpha(s)\boldsymbol{a} + \beta(s)\boldsymbol{x}_t$, where $\alpha(0) = \beta(1) = 1$, $\alpha(1) = \beta(0) = 0$ and $\alpha, \beta$ are smooth functions of $s$. Assume the density of $I_s(t)$ exists, is smooth in space, and denote it by $\rho_I(t, s)$. Then,*

$$v(\boldsymbol{x}, t, s) := \mathbb{E}[\partial_s I_s(\boldsymbol{a}, \boldsymbol{x}_t) | I_s(\boldsymbol{a}, \boldsymbol{x}_t) = \boldsymbol{x}]$$
$$= \frac{\partial_s \beta}{\beta} \boldsymbol{x} + \alpha^2 \left( \frac{\partial_s \beta}{\beta} - \frac{\partial_s \alpha}{\alpha} \right) \nabla \log \rho_I(\boldsymbol{x}, t, s).$$

We see that $v$ inherits regularity from $\rho(t)$ as well as the particular choice of stochastic interpolant used to construct the intermediate $\rho_I(t, s)$. We can refine this statement using a result from (Daniels, 2025): When $\rho(x, t) \propto \exp(-V(x, t))$, then

$$\rho_I(x, t, s) = \log \int e^{-V(y,t) - \frac{1}{2}\frac{\beta(s)^2}{\alpha(s)^2}|y|^2 + \frac{\beta(s)}{\alpha(s)^2}y \cdot x} \mathrm{d}y.$$

This formula makes it very explicit that the regularity of $v$, and in particular its derivative in $x$ and $t$, depends entirely on that of $t \mapsto \rho(t)$.

### 2.5. Relation to Minimal Energy (OT)

In general, the physics-time velocity field $u$ that we obtain via the two-parameter flow $\Phi$ is not the minimal kinetic

energy field that minimizes (2) over all admissible velocity fields that solve the continuity equation. We now discuss one specific case when we recover the minimal kinetic energy field and give an intuition that in general we do not.

**An example when our physics-time velocity is of minimal kinetic energy** Consider the Gaussian time marginals $\rho(t) = \mathcal{N}(0, \Sigma(t))$ with time-varying covariance matrix $\Sigma(t) \in \mathbb{R}^{d \times d}$. The base measure $\nu$ is standard normal. We derive the sampling-time velocity $v$ with stochastic interpolants and show that the induced physics-time velocity is linear and symmetric in $\boldsymbol{x}$, which means it has gradient form and thus minimizes the kinetic energy (2) among all admissible vector fields.

**Proposition 2.5.** *Let $I_s(t)$ be a stochastic interpolant from $\nu$ to $\rho(t)$ as in Proposition 2.4 and $v(\boldsymbol{x}, t, s) = \mathbb{E}[\partial_s I_s(\boldsymbol{a}, \boldsymbol{x}_t)|I_s(\boldsymbol{a}, \boldsymbol{x}_t) = \boldsymbol{x}]$. Suppose $\rho(t) = \mathcal{N}(0, \Sigma(t))$ is Gaussian. The corresponding physics-time velocity (7) is $u(\boldsymbol{x}, t, s) = U(t, s)\boldsymbol{x}$ with a matrix $U(t, s) \in \mathbb{R}^{d \times d}$. The matrix $U(t, s)$ is symmetric if and only if $\partial_t \Sigma(t)$ commutes with $\Sigma(t)$, in which case $u$ minimizes the kinetic energy (2) among all admissible velocity fields.*

**Discussion of minimal vs non-minimal kinetic energy fields** For example, consider a diagonal $\Sigma(t) = \mathrm{diag}(\sigma_1(t), \ldots, \sigma_d(t))$, which means that over time $t$ the Gaussian only "stretches" or "shrinks" but the principal directions (eigenvectors) remain the same. Such an evolution is compatible with pure gradient-field transport; no other dynamics are needed. Because $\partial_t \Sigma(t)$ is also diagonal in this case and diagonal matrices commute, the $u$ obtained as per Proposition 2.5 has minimal kinetic energy. In contrast, if the eigenvectors of $\Sigma(t)$ rotate with time $t$, then also the principal components of the distribution change over time $t$, which is a rotational evolution. In this setting, $\Sigma(t)$ and $\partial_t \Sigma(t)$ are not commuting anymore and our approach does not recover a minimal-kinetic energy field. Importantly, we do not view this as a limitation of the method. Rather, it reflects the presence of genuine rotational dynamics in the data. In such cases, enforcing a purely gradient (potential) structure can lead to highly irregular fields (Gigli (2011), Example 2.4), whereas our construction allows the learned field $u$ to represent the intrinsic rotation. In particular, this also means that our velocity $u$ can capture curl and rotations more naturally than a time-dependent gradient field can, as we will show in the numerical experiments.

**Enforcing properties of $u$** In some settings, it can be desirable to enforce certain constraints on the horizontal velocity $u$ that are, for example, informed by physical considerations. Since $u$ is determined entirely by $v$ through equation (10), these constraints have to be set on $v$ and ultimately come down to the choice of stochastic interpolant.

We give a concrete example: When $\nabla \cdot v = 0$ for all $x$, $t$, and $s$, then it follows that $\nabla \cdot u = 0$. This can be seen by taking the divergence of the compatibility condition, which, together with $\nabla \cdot v = 0$, implies

$$(\partial_s + v \cdot \nabla)(\nabla \cdot u) = 0 \Rightarrow \frac{\mathrm{d}}{\mathrm{d}s}(\nabla \cdot u)(\Phi_{t,s}(a)) = 0$$

for all $a$, $t$, and $s$. It follows that $\nabla \cdot u$ is preserved along the flow and therefore $(\nabla \cdot u)(x, t, s = 1) = (\nabla \cdot u)(\Phi_{t,s=1}^{-1}(x), t, s = 0) = 0$. This allows us to restrict the velocities learned by the TPF method to divergence-free fields.

## 3. Training and Inference

We now describe how to train a physics-time velocity field $u$ by first training a sampling-time velocity with standard conditional flow matching and then extracting $u$.

**Step 1: Learning the conditional flow matching model** Let us consider training data given in the form of samples from the time marginals. Consider the sets $\{\boldsymbol{x}_{t_k}^{(i)}\}_{i=1}^N \sim \rho(t_k)$ for time steps $k = 0, \ldots, K$ with $0 = t_0 < \cdots < t_K = T$, each containing $N$ samples per time marginal. These are independent samples at all times $t_0, \ldots, t_K$ and thus no trajectories in physics time are required. Furthermore, the number of time-marginal samples can differ across time steps.

We train a single conditional flow-matching model $v(\boldsymbol{x}, s, t; \theta_v)$ using samples $\{\boldsymbol{x}_{t_k}^{(i)}\}_{i=1}^N$ across all physics times $t_k$. For $k = 0, \ldots, K$, we then evaluate this model at $t = t_k$ to obtain the time-conditioned velocity fields $v(\cdot, \cdot, t_k) : \mathbb{R}^d \times [0, 1] \to \mathbb{R}^d$. Each conditional slice can be integrated in sampling time $s$ to (approximately) transport samples from the base distribution $\nu$ to the corresponding time marginal $\rho_{t_k}$.

**Step 2: Extracting physics-time velocity** Building on the velocities $v_0, \ldots, v_K$, we use them to generate $i = 1, \ldots, M$ trajectories $\hat{\boldsymbol{x}}_{t_0}^{(i)}, \hat{\boldsymbol{x}}_{t_1}^{(i)}, \ldots, \hat{\boldsymbol{x}}_{t_K}^{(i)}$ corresponding to the $M$ realizations $\boldsymbol{a}^{(1)}, \ldots, \boldsymbol{a}^{(M)}$ of samples of the base distribution $\nu$. Importantly, all states within a trajectory with index $i$ are obtained from the same noise realization $\boldsymbol{a}^{(i)}$ and thus the states within a trajectory are coupled across physics time, in contrast to the original independent samples from the marginals. In particular, if our theory from above applies with a $C^2$-diffeomorphism $\Phi$, then for each fixed $\boldsymbol{a} \sim \nu$, the map $t \mapsto \Phi(\boldsymbol{a}, t, 1)$ is continuously differentiable and therefore admits well-defined derivatives in time. Consequently, the synthetic trajectories can be seen as discretized samples generated with $u$.

We use the generated trajectories to estimate $u$ by fitting a (neural-network) parametrized velocity $u_\theta : \mathbb{R}^d \times [0, T] \to$

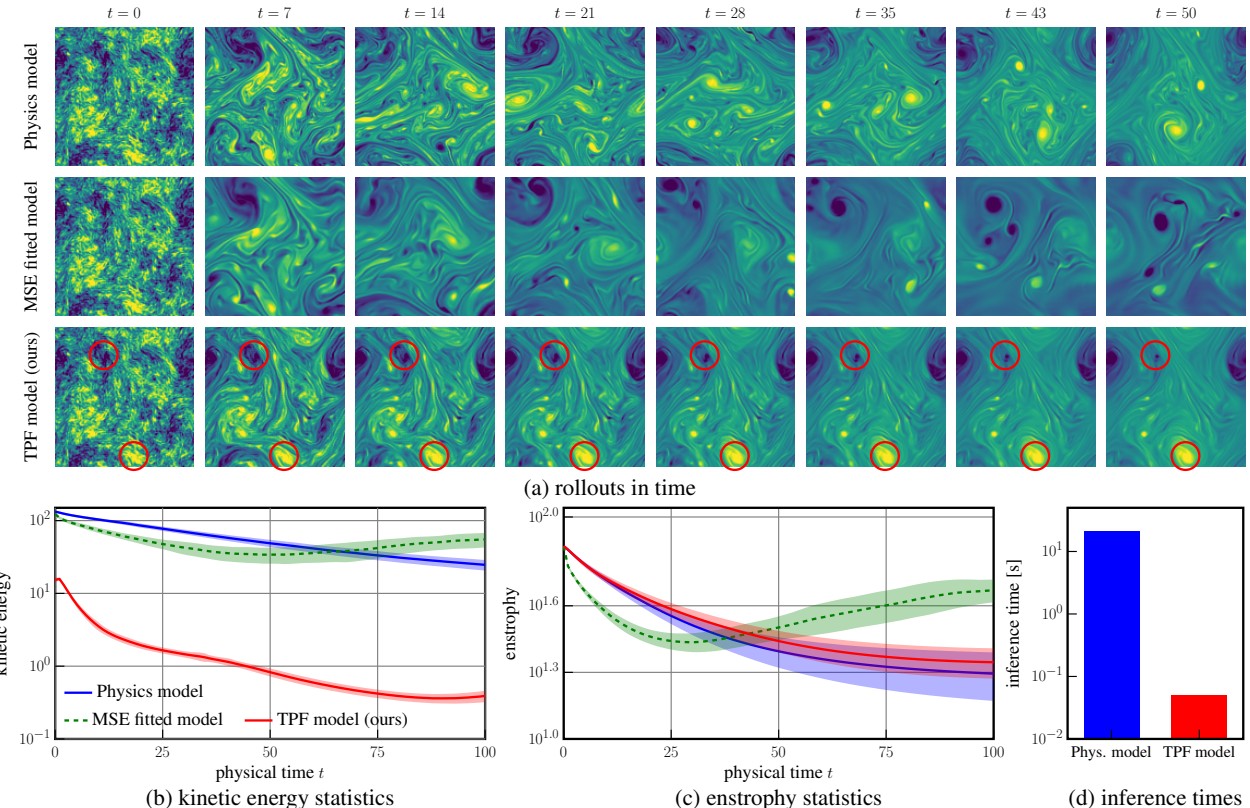

(a) rollouts in time

(b) kinetic energy statistics

(c) enstrophy statistics

(d) inference times

*Figure 3.* Barotropic flow: Our two-parameter flow approach scales population-dynamics inference to turbulent systems with states in $d > 10^4$ dimensions. It learns a fast-to-evaluate population-dynamics model that bypasses fine-scale advection motion (see lower kinetic energy of TPF model, which is emphasized in the trajectory videos found in the supplementary material) while preserving the self-organization/vortex-merging effects that are needed to match distributional quantities such as enstrophy. In contrast, pointwise mean-squared error (MSE) trajectory fitting targets individual sample paths and, in this chaotic regime, remains accurate only over short time horizons and fails to reliably reproduce population-level statistics.

$\mathbb{R}^d$ with a standard regression loss to the synthetic trajectories as

$$\min_\theta \sum_{i=1}^{M} \sum_{k=0}^{K-1} \left\| u_\theta(\hat{\boldsymbol{x}}_{t_k}^{(i)}, t_k) - \frac{\hat{\boldsymbol{x}}_{t_{k+1}}^{(i)} - \hat{\boldsymbol{x}}_{t_k}^{(i)}}{t_{k+1} - t_k} \right\|^2. \quad (11)$$

**Generating new samples (inference) and generalizing over physics parameters** Once we have $u_\theta$, we can integrate it with a new realization $\boldsymbol{x}_0$ of $\rho(0)$ as initial condition to generate a trajectory $\tilde{\boldsymbol{x}}_{t_1}, \ldots, \tilde{\boldsymbol{x}}_{t_K}$. Notice we only evolve in physics time when integrating with the learned $u_\theta$. In particular, with explicit Euler time-stepping, only a single network evaluation is needed per time step $k = 1, \ldots, K$.

We can also add conditioning on physics parameters that we denote by $\mu$. In this case, sampling-time velocity $v$ depends on $\boldsymbol{x}, t, s, \mu$, physics-time velocity $u$ at $s = 1$ depends on $\boldsymbol{x}, t, \mu$, and the loss (11) is extended with an empirical expectation over the training values of $\mu$.

**Summary of our procedure** We obtain our physics-time velocity field in three steps as follows.

1. **Train sampling-time transport.** Learn a conditional flow-matching model $\Phi(\boldsymbol{a}, t, s)$ that maps a base distribution $\nu$ to each time marginal $\rho(t)$.

2. **Generate coherent trajectories.** Sample $\boldsymbol{a}^{(i)} \sim \nu$ and define $\hat{\boldsymbol{x}}_{t_k}^{(i)} = \Phi(\boldsymbol{a}^{(i)}, t_k, 1)$ by integrating $v$ in sampling time $s$ for all $t_k$. This couples samples across physics time via a shared noise realization.

3. **Regress physics-time dynamics.** Fit $u_\theta(\boldsymbol{x}, t)$ to the finite-difference time derivatives of the synthetic trajectories using (11).

## 4. Experiments

**Evolving Gaussian mixture** Let us compare the velocities that we obtain with our two-parameter flows to optimal-transport/minimal-kinetic-energy velocities. Consider therefore a Gaussian mixture distribution $\rho(t)$ in $\mathbb{R}^2$ for which

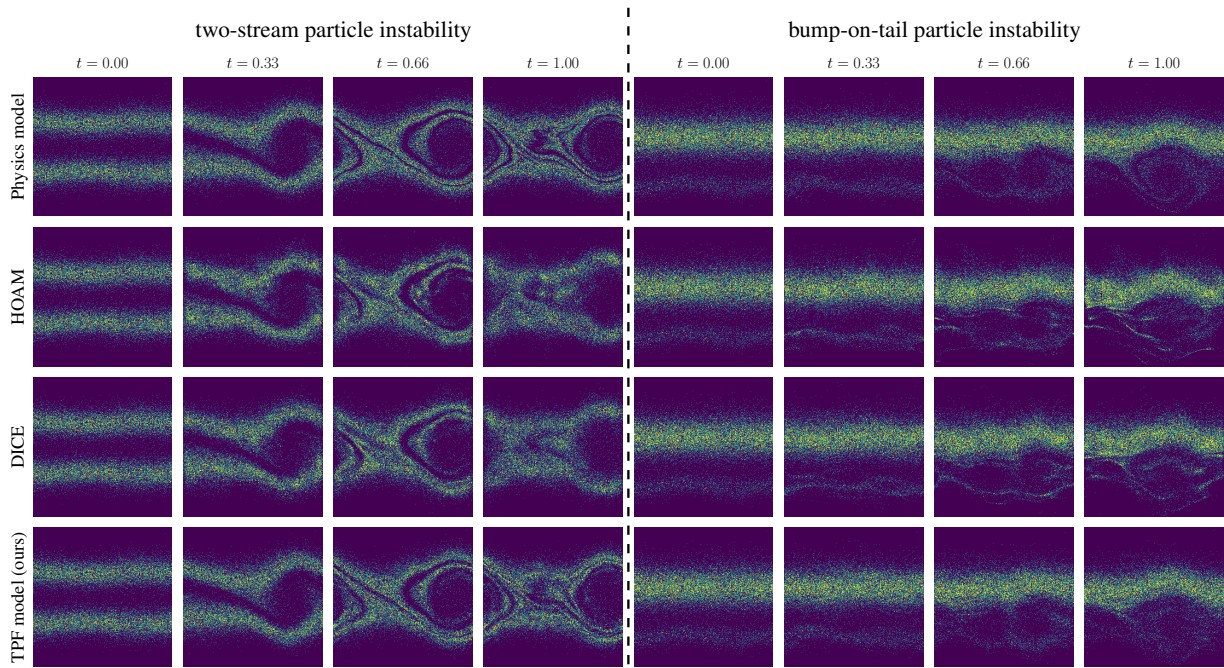

*Figure 4.* Particle instabilities: Our TPF method learns population dynamics that accurately generalize across unseen Debye-length parameters in these Vlasov-Poisson instability benchmarks and match fine phase-space structure.

means and covariances undergo a random walk over $t$. We train a two-parameter flow model as well as a flow with optimal-transport couplings between successive time steps following Terpin et al. (2024); see Appendix A.1. Figure 2 tracks samples over time and color-codes their position, which shows that our approach leads to more regular dynamics than induced by the optimal-transport couplings. Note that this is in agreement with our discussion in Section 1.4.

**Barotropic flow**  To demonstrate the effectiveness of our approach on high-dimensional distributions, we study data from the time marginals generated from a vorticity transport equation. We set the initial vorticity as a Gaussian random field and then obtain the vorticity on a $d = 128 \times 128$ grid as samples over time. It is a classic result (McWilliams, 1984) that such randomly initialized vorticity fields self-organize into isolated coherent vortices (see top row in Figure 3). However, resolving this evolution requires small time steps due to the Courant-Friedrichs-Lewy (CFL) condition, and the resulting dynamics are dominated by rapid advective transport and vortex rotation. For our purposes of computing population-level quantities of interest, we seek population-level dynamics that capture the slow vortex-merging process without reproducing the full fine-scale transport.

***Results*** Our two-parameter flow model recovers precisely the self-organization into isolated coherent vortices (bottom row of Fig. 3(a)). The dominant vortices remain approximately stationary while the vorticity field gradually

organizes into fewer and larger coherent structures with substantially less motion and thus less kinetic energy. As shown in Fig. 3(b), the kinetic energy of the inferred population dynamics is roughly two orders of magnitude lower than that of the physical trajectories, indicating that fast advective motion has been effectively removed while the population-level effect (the aggregation of vorticity into coherent vortices) is preserved. This is further confirmed by the population enstrophy in Fig. 3(c), where the temporal decay of enstrophy closely matches that of the true dynamics. Additionally, because on the population level we do not have to resolve fast advective motion, we can take larger time steps and achieve orders of magnitude speedup in runtime (inference time) compared to using the numerical simulation with the physics model (Fig. 3(d)). Note that an MSE-fitted model to the trajectories aims to capture the fast advective motion but ultimately fails to capture the vortex-merging behavior and so leads to a wide mismatch in the enstrophy.

***TPF scales population-dynamics inference to high dimensions*** We emphasize that this example at dimension $d = 128^2$ is a significant step up compared to many other existing works on population dynamics inference. The largest dimensions considered in previous work we are aware of are $32^2$ (Neklyudov et al., 2023; Blickhan et al., 2025), 100 (Chen et al., 2023; Persiianov et al., 2026), and 50 (Terpin et al., 2024; Petrović et al., 2025).

*Table 1.* Kolmogorov flow: Samples generated with TPF match the energy spectrum decay $\omega^{-3}$ closely.

| time step | phys. model | MSE-fitted | TPF (ours) |
|---|---|---|---|
| 32 | $\omega^{-2.800}$ | $\omega^{-4.164}$ | $\omega^{-2.712}$ |
| 64 | $\omega^{-2.879}$ | $\omega^{-3.928}$ | $\omega^{-2.894}$ |
| 95 | $\omega^{-2.932}$ | $\omega^{-3.818}$ | $\omega^{-2.890}$ |
| 127 | $\omega^{-2.970}$ | $\omega^{-3.805}$ | $\omega^{-2.880}$ |

*Table 2.* Particle instabilities: TPF achieves competitive accuracy compared to other population-dynamics inference approaches.

| Method | two-stream | | bump-on-tail | |
|---|---|---|---|---|
| | avg. ($\downarrow$) | final ($\downarrow$) | avg. ($\downarrow$) | final ($\downarrow$) |
| DICE | 4.5e-04 | 7.8e-04 | 6.5e-03 | 1.6e-02 |
| HOAM | 2.8e-03 | 4.0e-03 | 3.8e+01 | 5.5e+01 |
| AM | 4.6e+00 | 1.1e+00 | 2.5e-03 | 4.5e-03 |
| JKONet* | 7.7e-03 | 9.8e-03 | 4.3e-03 | 7.5e-03 |
| TPF (ours) | **3.2e-04** | **4.0e-04** | **3.1e-04** | **4.7e-04** |

**Kolmogorov flow**   Let us now consider the Kolmogorov flow following (Kochkov et al., 2021). We directly use the code coming with (Kochkov et al., 2021) to generate training data, analogous to the setup for the barotropic flow; see Section A.3. We then train a TPF model and an MSE-fitted (operator learning) model analogous to the previous section. In this example, the energy spectrum should decay as $\omega^{-3}$, where $\omega$ is the frequency (wave number). We obtain decay rates shown in Table 1 when fitting to the spectrum decay of samples generated with the physics model, an MSE-fitted (operator learning) model, and TPF. Our TPF matches the exponent $-3$ closely (within estimation error when comparing to the exponent estimated from the data generated with the physics model). These results show that the TPF generated samples are physically meaningful in the sense that they capture properties that depend on the distribution of the states.

**Vlasov-Poisson instabilities**   Let us now consider particles governed by the Vlasov-Poisson system. We investigate the two-stream and bump-on-tail instabilities in two-dimensional phase space with $N = 25 \times 10^3$ samples. We adopt the experimental configuration described in (Berman et al., 2024, Section B.2). Specifically, the system is parametrized by a characteristic (Debye) length parameter, denoted by $\mu$.

***Results***   We report the averaged $W_2$ error over test characteristic length parameters $\mu$ in Table 2; see Appendix A.4 for details. We compare to action matching (AM) (Neklyudov et al., 2023), HOAM (Berman et al., 2024), DICE (Blickhan et al., 2025), and JKONet* (Terpin et al., 2024), which are alternative methods to learn population dynamics; see Introduction. Density plots of the phase-space density are shown in Figure 4. We see that the HOAM (Berman et al., 2024) and DICE (Blickhan et al., 2025) methods, which are based on the ansatz $u = \nabla\varphi$, do not manage to capture the fine structures of the phase-space filamentation as $t \approx 1$. As we mentioned earlier, training with the ansatz $u = \nabla\varphi$ guarantees to obtain only gradient dynamics but it can be challenging to capture rotational dynamics. As the vortices in the two-stream and bump-on-tail instability rotate, our approach that is not tied to gradient dynamics can achieve higher accuracy.

## 5. Conclusions and Limitations

*Conclusions*   We present two-parameter flows as a practical and principled route to population dynamics inference. We learn scalable base-to-marginal transports with standard conditional flow methods and then extract physics-time velocity fields via regression for fast generation of samples over physics time. In particular, we show that the physics-time dynamics are well-posed and well-behaved as long as the base-to-marginal velocities are regular. Empirically, this yields a robust and efficient population-dynamics-inference method for regimes where OT (and JKO-style) approaches are computationally prohibitive or ill-suited, including high-dimensional problems and rotational dynamics.

*Limitations*   The inferred physics-time dynamics inherit inductive bias from the chosen base-to-marginal transport. Beyond matching marginals and the regularity bounds that we provided, future work is needed to understand what additional structure on the base-to-transport velocity is needed for the induced physics-time velocity to satisfy further criteria such as low kinetic energy, minimal curl, a low Lipschitz constant, or interpretability (see also the discussion at the end of Section 2.5).

## Acknowledgments

This material is based upon work supported by the U.S. Department of Energy, Office of Science under Award #DE-SC0024721. The authors have been funded in part by the Air Force Office of Scientific Research (AFOSR), USA, award FA9550-24-1-0327.

## Impact Statement

This paper presents work whose goal is to advance the field of Machine Learning. There are many potential societal consequences of our work, none which we feel must be specifically highlighted here.

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

# A. Details about examples

## A.1. Evolving mixture of Gaussians

A conditional flow matching model learns velocity fields to map a standard Gaussian $\mathcal{N}(0, I)$ to each marginal $\rho(t)$, trained on linear interpolation paths $\boldsymbol{x}_s = (1 - s)\boldsymbol{a} + s\boldsymbol{x}_1$. The marginals are constructed by placing a mixture of five Gaussians with random mean values $\sim \mathcal{N}(0, 1.5)$ and covariances $\sim \mathrm{Unif}([0.1, 0.4])$. Over time, the mean values perform a random walk (step size $= 0.7$) and the covariances perform a random walk on the space $2 \times 2$ matrices, with a projection to the set of positive semidefinite matrices after every step. On top of this, there is a rotational field $0.25[-x^2, x^1]$ and an inward drift $-0.1x$ acting on every sample. We generate six time marginals with this procedure with $N = 3000$ samples each.

We compare the trajectories inferred by TPF to those obtained by connecting adjacent time marginals with OT maps. Just as in (Terpin et al., 2024), we can compute the OT couplings exactly using the Hungarian algorithm due to the relatively low number of samples $N$. The results are visualized in Figure 2 by coloring all particles at $t = 0$ (the color of the sample at $\boldsymbol{x}^{(i)}$ is based on the radial coordinate of the noise sample $\boldsymbol{a}^{(i)} = \Phi^{-1}(\boldsymbol{x}^{(i)}, t, 1)$) and keeping this color constant with $t$.

## A.2. Barotropic flow

The two-dimensional vorticity transport equation is $\partial_t \omega = -\boldsymbol{u} \cdot \nabla \omega + \nu \Delta \omega + \kappa \Delta^4 \omega$, posed on the periodic domain $[-\pi, \pi)^2$. The velocity field $\boldsymbol{u} = (u_1, u_2)^\top$ is obtained from the stream function $\Psi(\boldsymbol{x})$ via

$$-\Delta \Psi = \omega, \qquad u_1 = \partial_y \Psi, \qquad u_2 = -\partial_x \Psi,$$

so that the state of the system is fully determined by the vorticity $\omega$. To study the inviscid regime, we set $\nu = 0$ and use a hyperviscosity coefficient $\kappa = 10^{-2}$ to ensure numerical stability. We discretize the domain using a Fourier spectral method with $128 \times 128$ modes and draw the initial vorticity field from a Gaussian random field. Thus, the dimension of our data points is $d = 128^2$. We generate 3000 trajectories with different initializations as training data for conditional flow matching.

The kinetic energy ("field difference") of the sample ensemble is computed as $\frac{1}{2} \sum_{j=1}^{K} \sum_{i=1}^{N} |\boldsymbol{x}_{t_{j+1}}^{(i)} - \boldsymbol{x}_{t_j}^{(i)}|^2 / |t_{j+1} - t_j|$ with $\boldsymbol{x} \in \mathbb{R}^{128^2}$, where $\boldsymbol{x}_t$ represents a vorticity field $\omega$ on a $128 \times 128$ grid.

We parameterize the flow matching velocity field with a convolutional U-Net architecture. The model is trained for 400 epochs with a learning rate of $10^{-4}$ using the AdamW optimizer. The U-Net uses a base channel width of 32 and a six-level encoder-decoder structure with skips with channel multipliers $(1, 2, 2, 4, 4, 8)$ and strides $(1, 2, 1, 2, 1, 2)$. Physics time and sampling time are embedded through an MLP with embedding dimension 32 and hidden width 64 and injected into the network. We generate 1024 coherent trajectories for the regression model. The regression models (both MSE and TPF) use the same architecture but only train for 200 epochs (note that these converge much quicker than the flow matching model). The code to recreate the experimental results is available at `https://github.com/Algopaul/tpf`.

## A.3. Kolmogorov flow

We use the setup following (Kochkov et al., 2021) and described and implemented in `https://github.com/google/jax-cfd` under "forced turbulence." The neural network and training setup is analogous to Section A.2.

## A.4. Vlasov-Poisson instabilities

The dynamics of a particle at phase-space position $\boldsymbol{x}_t = [x_t^1, x_t^2]^\top$ are given by

$$\frac{\mathrm{d}}{\mathrm{d}t} \begin{bmatrix} x_t^1 \\ x_t^2 \end{bmatrix} = \begin{bmatrix} x_t^2 \\ \nabla \phi(t, x^1) \end{bmatrix},$$

where the potential $\phi$ satisfies the Poisson equation:

$$-\mu^2 \Delta \phi = 1 - \int \rho(t, \boldsymbol{x}, \mu) \, \mathrm{d}x^2,$$

where $\rho$ denotes the phase-space density.

The parameter $\mu$ is selected from the sets $\mu_{\text{train}} \in \{1.2, 1.3, \ldots, 1.9\}$ and $\mu_{\text{test}} \in \{1.25, 1.85\}$. The evaluation metric we consider is the $W_2$ distance between the law of the test data $\boldsymbol{x}_t(\mu_{\text{test}}) \sim \rho(\cdot, t, \mu_{\text{test}})$ and the data generated by the population dynamics learned by the two-parameter flow, $\rho^{TPF}$:

$$\Delta(t, \mu) := W_2(\rho, \rho^{TPF})(t, \mu).$$

In Table 2, we report $\frac{1}{3} \sum_{t \in 0.33, 0.66, 1.0} \Delta(t, \mu)$ and the final $\Delta(t, \mu)$ on test data.

For the conditional flow-matching model we use a multilayer perceptron (MLP) parameterization of the velocity field. Training is performed for 1000 epochs with a batch size of $10^5$ particles (note that this high batch size is possible since the dimension of each sample is 2) and a learning rate of $10^{-4}$ using the AdamW optimizer. The vector field is represented by a 6-layer MLP with 256 hidden features per layer, GELU activations, residual (skip) connections, and no batch normalization. The network outputs a two-dimensional velocity field. Experiments are run on an NVIDIA L40S GPU. We generate 60 coherent trajectories at linearly spaced parameters $\mu$ for the regression model. The regression model has the same architecture but is trained with learning rate $10^{-3}$. The code to recreate the experimental results is available at https://github.com/Algopaul/tpf.

# B. Proofs

## B.1. Compatibility condition

**Proposition B.1.** *For $\Phi$ of class $C^2$ with velocity fields $u : \partial_t \Phi = u \circ \Phi$ and $v : \partial_s \Phi = v \circ \Phi$, it holds that $\partial_s u + v \cdot \nabla u - u \cdot \nabla v - \partial_t v = 0$.*

*Proof.* For twice differentiable $\Phi$, the mixed derivatives with respect to $t$ and $s$ commute as a consequence of Schwarz' theorem: $(\partial_t \partial_s - \partial_s \partial_t) \Phi(\boldsymbol{a}, t, s) \equiv 0 \; \forall (\boldsymbol{a}, t, s)$. Expanding this expression gives, with $\boldsymbol{x}_{t,s} = \Phi(\boldsymbol{a}, t, s)$ for short,

$$\frac{\mathrm{d}}{\mathrm{d}t} \frac{\mathrm{d}}{\mathrm{d}s} \Phi(\boldsymbol{a}, t, s) = \frac{\mathrm{d}}{\mathrm{d}t} v(\boldsymbol{x}_{t,s}, t, s) = \partial_t v(\boldsymbol{x}_{t,s}, t, s) + \frac{d\Phi}{dt}(\boldsymbol{a}, t, s) \cdot \nabla v(\boldsymbol{x}_{t,s}, t, s) = (\partial_t v + u \cdot \nabla v)(\boldsymbol{x}_{t,s}, t, s).$$

The computation for $\frac{\mathrm{d}}{\mathrm{d}s} \frac{\mathrm{d}}{\mathrm{d}t} \Phi(\boldsymbol{a}, t, s)$ proceeds analogously and equating the two yields $(\partial_s u + v \cdot \nabla u - u \cdot \nabla v - \partial_t v)(\boldsymbol{x}_{t,s}, t, s) = 0$. Since there exists for all $\boldsymbol{x}$ a point $\boldsymbol{a}$ such that $\Phi(\boldsymbol{a}, t, s) = \boldsymbol{x}$, the proof is complete. $\square$

## B.2. Proof of Proposition 2.1

*Proof of Proposition 2.1.* For regularity and uniqueness of the flow, see (Abraham et al., 1988, Lemma 4.1.9) and (Abraham et al., 1988, Theorem 4.1.11). We now show that $u$ satisfies the continuity equation at sampling time $s = 1$.

The relation $\Phi(\cdot, t, 1) \sharp \nu = \rho(\cdot, t)$ implies $\langle \phi, \rho(\cdot, t) \rangle = \langle \phi \circ \Phi(\cdot, t, 1), \nu \rangle$ for any test function $\phi$, where $\langle \phi, \rho \rangle = \int \phi(x) d\rho(\boldsymbol{x})$. Differentiate in time to find

$$\langle \phi, \partial_t \rho(\cdot, t) \rangle = \langle \partial_t \Phi(\cdot, t, 1) \cdot (\nabla \phi \circ \Phi(\cdot, t, 1)), \nu \rangle = \langle u(\cdot, t, 1) \cdot \nabla \phi, \rho(\cdot, t) \rangle,$$

which is the weak form of the continuity equation. $\square$

## B.3. Proof of Proposition 2.2

*Proof of Proposition 2.2. Analytic form of $u$:* Let $u(\boldsymbol{x}_{t,s}, t, s) = D\Phi(\boldsymbol{a}, t, s) \cdot \int_0^s [D\Phi(\boldsymbol{a}, t, \sigma)]^{-1} \partial_t v(\boldsymbol{x}_{t,\sigma}, t, \sigma) \, d\sigma$.

Take derivative in s:

$$\frac{\mathrm{d}}{\mathrm{d}s} u(\boldsymbol{x}_{t,s}, t, s) = (\partial_s u + v \cdot \nabla u)(\boldsymbol{x}_{t,s}, t, s)$$

For the right-hand side, we use:

$$\frac{\mathrm{d}}{\mathrm{d}s} D\Phi(\boldsymbol{a}, t, s) = Dv(\boldsymbol{x}_{t,s}, t, s) D\Phi(\boldsymbol{a}, t, s)$$

and $\frac{\mathrm{d}}{\mathrm{d}s} \int_0^s f(\sigma) \, \mathrm{d}\sigma = f(s) \; \forall f$. Putting it all together, we find

$$\frac{\mathrm{d}}{\mathrm{d}s} u(\boldsymbol{x}_{t,s}, t, s) = Dv(\boldsymbol{x}_{t,s}, t, s) u(\boldsymbol{x}_{t,s}, t, s) + \underbrace{D\Phi(\boldsymbol{a}, t, s) \left[ D\Phi(\boldsymbol{a}, t, s) \right]^{-1}}_{= \mathrm{I}} \partial_t v(\boldsymbol{x}_{t,s}, t, s)$$

and hence

$$\partial_s u + v \cdot \nabla u = u \cdot \nabla v + \partial_t v$$

evaluated at $(\boldsymbol{x}_{t,s}, t, s)$. This relation holds for all $\boldsymbol{x}_{t,s}$, in particular we can find for all $\boldsymbol{x}$ an $\boldsymbol{a}$ such that $\boldsymbol{x} = \boldsymbol{x}_{t,s} = \Phi(\boldsymbol{a}, t, s)$ at times $(t, s)$.

We have shown that $u$ satisfies the compatibility condition. It remains to show that this implies that $u \circ \Phi = \partial_t \Phi$. By using $(\partial_t \partial_s - \partial_s \partial_t)\Phi = 0$, we see that

$$\partial_s \left( \partial_t \Phi(\boldsymbol{a}, t, s) \right) = \partial_t \partial_s \Phi(\boldsymbol{a}, t, s) = (\partial_t v + u \cdot \nabla v)(\Phi(\boldsymbol{a}, t, s)) = (\partial_s u + v \cdot \nabla u)(\Phi(\boldsymbol{a}, t, s)),$$

using the compatibility condition in the last equality. At the same time,

$$\frac{\mathrm{d}}{\mathrm{d}s} u(\Phi(\boldsymbol{a}, t, s)), t, s) = (\partial_s u + v \cdot \nabla u)(\Phi(\boldsymbol{a}, t, s)).$$

Hence, $u \circ \Phi$ and $\partial_t \Phi$ satisfy the same linear ODE and $\frac{\mathrm{d}}{\mathrm{d}s} (u \circ \Phi - \partial_t \Phi) = 0$. Since $\Phi(\boldsymbol{a}, t, 0) = \boldsymbol{a}$ implies $\partial_t \Phi \big|_{s=0} = 0$ and $u$ (defined in (10)) $= 0$ at $s = 0$, we conclude $u \circ \Phi = \partial_t \Phi$ on every integral curve of $\Phi$, i.e. at every $\boldsymbol{x} = \Phi(\boldsymbol{a}, t, s)$.

*Jacobian of $u$:* Denote $\boldsymbol{x}_{t,s} = \Phi(\boldsymbol{a}, t, s)$. By regularity of $\Phi$,

$$0 = \left( D_{\boldsymbol{a}} \frac{\mathrm{d}}{\mathrm{d}t} - \frac{\mathrm{d}}{\mathrm{d}t} D_{\boldsymbol{a}} \right) \Phi(\boldsymbol{a}, t, s) = D_{\boldsymbol{a}} u(\boldsymbol{x}_{t,s}, t, s) - \frac{\mathrm{d}}{\mathrm{d}t} D\Phi(\boldsymbol{a}, t, s) = (Du)(\boldsymbol{x}_{t,s}, t, s) D\Phi(\boldsymbol{a}, t, s) - \frac{\mathrm{d}}{\mathrm{d}t} D\Phi(\boldsymbol{a}, t, s).$$

$\square$

### B.4. Proof of Proposition 2.3

*Proof of Proposition 2.3.* First, note that $\langle f \circ \Phi(\cdot, t, s), \nu \rangle = \langle f, \rho_I(t, s) \rangle$ for all $f$ by definition of the push-forward condition $\Phi(\cdot, t, s) \sharp \nu = \rho_I(\cdot, t, s)$. We now compute (omitting the dependence on $(t, s)$ for the sake of brevity):

$$\frac{1}{2} \frac{\mathrm{d}}{\mathrm{d}s} \|u \circ \Phi\|_{L^2(\nu)}^2 = \langle (u \cdot (\partial_s u + v \cdot \nabla u)) \circ \Phi, \nu \rangle = \langle u \cdot (\partial_t v + u \cdot \nabla v), \rho \rangle \leq \|\partial_t v\|_{L^2(\rho)} \|u\|_{L^2(\rho)} + \|\nabla v\|_\infty \|u\|_{L^2(\rho)}^2.$$

We apply here, in order, the total derivative of $u$ with respect to $s$, the compatibility condition, the change of variables from $\nu$ to $\rho$, and Hölder's inequality. $\| \dots \|_\infty$ denotes the maximum norm.

Completely analogously, we compute

$$\frac{1}{2} \frac{\mathrm{d}}{\mathrm{d}s} \|\nabla u \circ \Phi\|_{L^2(\nu)}^2 = \langle \nabla u : (\partial_t \nabla v - (\nabla u) \cdot \nabla v + (\nabla v) \cdot \nabla u + u \cdot \nabla^2 v), \rho \rangle,$$

where $\nabla^2 v$ denotes the tensor with entries $\partial_{x_i} \partial_{x_j} v_k$ and $:$ denotes the Frobenius inner product between matrices.

All together, this implies

$$\frac{1}{2} \frac{\mathrm{d}}{\mathrm{d}s} \|u\|_{H^1(\rho)}^2 \leq \|\partial_t v\|_{L^2(\rho)} \|u\|_{L^2(\rho)} + \|\nabla v\|_\infty \|u\|_{L^2(\rho)}^2 + \|\partial_t \nabla v\|_{L^2(\rho)} \|\nabla u\|_{L^2(\rho)}$$
$$+ 2\|\nabla v\|_\infty \|\nabla u\|_{L^2(\rho)}^2 + \|\nabla^2 v\|_\infty \|\nabla u\|_{L^2(\rho)} \|u\|_{L^2(\rho)}$$

Next, use the Cauchy-Schwarz inequality to obtain

$$\|\partial_t v\|_{L^2(\rho)} \|u\|_{L^2(\rho)} + \|\partial_t \nabla v\|_{L^2(\rho)} \|\nabla u\|_{L^2(\rho)} \leq \sqrt{\|\partial_t v\|_{L^2(\rho)}^2 + \|\partial_t \nabla v\|_{L^2(\rho)}^2} \sqrt{\|u\|_{L^2(\rho)}^2 + \|\nabla u\|_{L^2(\rho)}^2}$$

and Young's inequality for

$$\|\nabla^2 v\|_\infty \|\nabla u\|_{L^2(\rho)} \|u\|_{L^2(\rho)} \le \frac{1}{2}\|\nabla^2 v\|_\infty \left(\|u\|_{L^2(\rho)}^2 + \|\nabla u\|_{L^2(\rho)}^2\right) = \frac{1}{2}\|\nabla^2 v\|_\infty \|u\|_{H^1(\rho)}^2$$

to simplify to

$$\frac{1}{2}\frac{\mathrm{d}}{\mathrm{d}s}\|u\|_{H^1(\rho)}^2 = \|u\|_{H^1(\rho)}\frac{\mathrm{d}}{\mathrm{d}s}\|u\|_{H^1(\rho)} \le \|\partial_t v\|_{H^1(\rho)}\|u\|_{H^1(\rho)} + \left(\frac{1}{2}\|\nabla^2 v\|_\infty + 2\|\nabla v\|_\infty\right)\|u\|_{H^1(\rho)}^2.$$

Divide by $\|u\|_{H^1(\rho)}$ and apply Gronwall's Lemma with $u\big|_{s=0} \equiv 0$ to find

$$\|u\|_{H^1(\rho)}(t,1) \le \int_0^1 \|\partial_t v\|_{H^1(\rho)}(t,s) \exp\left(\int_s^1 \left(\frac{1}{2}\|\nabla^2 v\|_\infty + 2\|\nabla v\|_\infty\right)(t,\sigma)\,\mathrm{d}\sigma\right)\mathrm{d}s$$

as claimed. $\qquad\square$

### B.5. Proof of Proposition 2.5

*Proof of Proposition 2.5.* Note that we are interested in the case with $s = 1$. *Step 1:* First, let us state that

$$v(\boldsymbol{x},t,s) = \frac{1}{2}\partial_s \log M(t,s)\boldsymbol{x},$$

with matrices $M(t,s) = \alpha(s)^2 + \beta(s)^2\Sigma(t)$ (note that $\partial_s M$ commutes with $M$). The matrix corresponding to the velocity $u$ is therefore $U(t,s) = (\partial_t M(t,s)^{1/2})M(t,s)^{-1/2}$. The formula for $v$ is standard, see for example (Chen et al., 2025, Proposition 3.9). We can integrate this flow to obtain

$$\Phi(\boldsymbol{a},t,s) = M(t,s)^{1/2}\boldsymbol{a}.$$

We now compute $u$.

$$\frac{d\Phi}{dt}(\boldsymbol{a},t,s) = \partial_t M(t,s)^{1/2}\boldsymbol{a} = \left(\partial_t M(t,s)^{1/2}\right)M(t,s)^{-1/2}\Phi(\boldsymbol{a},t,s),$$

hence

$$u(\boldsymbol{x},t,s) = \left(\partial_t M(t,s)^{1/2}\right)M(t,s)^{-1/2}\boldsymbol{x}.$$

To obtain the expression for $U$, note that $\partial_t(M^{1/2}M^{1/2}) = (\partial_t M^{1/2})M^{1/2} + M^{1/2}\partial_t(M^{1/2}) = \partial_t M = \beta^2\partial_t\Sigma$, where $(\partial_t M^{1/2})M^{1/2} = (\partial_t M^{1/2})M^{-1/2}M = UM$ and $M^{1/2}\partial_t M^{1/2} = M^{1/2}UM^{1/2}$. In particular, $U(t,s)$ solves the Sylvester equation

$$UM + M^{1/2}UM^{1/2} = \beta^2\partial_t\Sigma.$$

*Step 2:* We have $U - U^T = (\partial_t M^{1/2})M^{-1/2} - M^{-1/2}\partial_t M^{1/2}$ since $M$ is symmetric. Multiply by (the full-rank) $M^{1/2}$ from the left and right to obtain $M^{1/2}\partial_t M^{1/2} - (\partial_t M^{1/2})M^{1/2} = [M^{1/2},\partial_t M^{1/2}]$. Since $[M^{1/2},\partial_t M^{1/2}] = 0 \Leftrightarrow [M,\partial_t M] = 0 \Leftrightarrow [\Sigma,\partial_t\Sigma] = 0$, we find $U = U^T \Leftrightarrow [\Sigma,\partial_t\Sigma] = 0$. The fact that $M^{1/2}$ commutes with its derivative if and only if $M$ does follows from the fact that the two are simultaneously diagonalizable, see for example (Higham, 2008, Chapter 3).

When $U$ is symmetric, $u(\boldsymbol{x},t,s) = \frac{1}{2}\nabla(\boldsymbol{x}^T U(t,s)\boldsymbol{x})$ and the admissible gradient field is kinetic energy-optimal (Ambrosio et al., 2005, Proposition 8.4.3). $\qquad\square$

