# OpenReview forum: "Two-Parameter Flows for Learning Population Dynamics of Physical Systems"
_ICML.cc/2026/Conference — ICML 2026 regular_

### Official Review · Reviewer_zGm8 · 2026-03-04

**Soundness:** 3
**Presentation:** 3
**Significance:** 2
**Originality:** 2
**Overall Recommendation:** 4
**Confidence:** 4

**Summary:**

The authors propose a novel approach to learn the population dynamics of a physical system where training data consists of independent samples at different time points. Instead of relying on particle trajectories, the method learns hot the density evolves over physical time. To do so they train a conditional flow matching model that maps a base distribution to the distributions at all time points. This part is conditioned on a sampling (not physical) time. However, by means of a regression step it is possible to extract a velocity in the physical time space. The underlying assumption is that if the conditional flow matching model is sufficiently regular, then also the dynamics in physical time will be regular. However, in general there are many velocity fields that explain the same dynamics, as long as they satisfy the continuity equation.

**Compliance With Llm Reviewing Policy:**

Affirmed.

**Final Justification:**

Within the rebuttal the authors included more experiments which helped contextualize the work better. My initial concerns about the overall relevance, which goes beyond this single paper, seem to be addressed by a scaling argument (at least in theory). This convinced me to increase my score to Weak accept.

**Key Questions For Authors:**

### Questions
1. How would the proposed model differ from directly learning a single time-conditioned diffeomorphism, i.e. a normalizing flow conditioned on time? also in that case one wouldn't need explicit trajectories and, after training, would be able to access the velocity field without a regression step (and also to the learnt trajectories, if needed) [1]. Relevantly, the evolution of the density in the physical time fulfills the continuity equation by construction. The reasoning behind the model is tha same as the one outlined in lines 187-191 right column. A slightly similar idea was also proposed in [2] but in a different context from the proposed approach, requiring initial conditions rather than observations.
2. If the presented approach differs significantly from the mentioned conditional normalizing flows, it would still be beneficial to include an experimental comparison.
3. How do HOAM and DICE perform on the barotropic flow experiment?
4. I am a bit puzzled about the comparison with the physics model in terms of kinetic energy (Figure 3b). Shouldn't the physical system be taken as the ground truth, like in Figure 3a? At first glance I thought this constituted a violation of the underlying physics
5. Does the method require the same number $N$ of observations for each time step?

### Typos
line 83 (right): "his" should be "is"

line 100 (left), 124 (left), 131-134 (left), 157(left), 161 (left), 114 (right) 414 (left): the citations should be part of the sentence (not in parenthesis)

line 269 (left) unfinished sentence

[1] F. Arend Torres et al., Lagrangian Flow Networks for Conservation Laws, ICLR 2024

[2] L. Li et al., Self-Consistent Velocity Matching of Probability Flows, NeuriPS 2023

**Limitations:**

Yes

**Strengths And Weaknesses:**

### Strengths
1. **well-motivated and described**: the proposed approach is well motivated and clearly explained. Statements are precise and rigorously justified.
2. **experimental validation**: experimental results show improved performance both in terms of quality and inference runtime, at least compared to numerical simulations.

### Weaknesses
1. **novelty**: the model essentially relies on conditional flow matching (or equivalent) and connects the dots concerning the evolution over physical time. If I understand correctly, this connection was already established for other conditional density models like Normalizing Flows. In particular, by training a conditional normalizing flows on the data points, it is possible to automatically learn an explicit velocity field in the physical space (by means of a similar reasoning as that of lines 187-191 right columns). I discuss this point more precisely in the questions section.
2. **comparison with other ML models**: the experimental comparison with othermML approaches is rather limited (one experiment out of three) and other comparisons are rather qualitative (Figure 2 and Figure 3). From a practical perspective, it is hard to place the proposed approach within the literature in terms of performance.

---

> ### Author Rebuttal · Authors · 2026-03-30
>
> We thank the reviewer for a careful and detailed reading of the paper.
>
> ***Relation to Normalizing Flows and [1,2]***
>
> A time-conditioned normalizing flow would directly learn a full physical-time diffeomorphism; our contribution is to show that this is unnecessary. Learning only (very) scalable base-to-marginal transport already suffices to recover a unique, regular, explicit physics-time dynamics model. In other words, we do less than directly learning a full time-conditioned diffeomorphism, yet still recover an explicit dynamics model in physical time.
>
> This difference matters because our goal is not merely to show that a velocity field exists, but to obtain a scalable population-dynamics pipeline. Our method deliberately uses conditional flow matching as a base-to-marginal backbone because it is an established scalable generative primitive, and then converts it into a cheap explicit physics-time model. E.g., the papers [1,2] cited by the reviewer consider dimensions up to $d=3$ and $d=60$, respectively. Our barotropic example is in dimension $d = 16,384$.
>
> ***A slightly similar idea was also proposed in [2] but in a different context from the proposed approach [...].***
>
> We thank the reviewer for pointing out [1,2] and we will cite them in a revision but their setting is fundamentally different from ours in two ways: scale (see previous answer), and more importantly, the state variable whose population dynamics is being learned. In [1,2], the state is a particle position in low-dimensional physical or phase space, so the learned law describes an ensemble of particles; in our barotropic example, a single sample $x \in \mathbb{R}^{16384}$ is an entire vorticity field on a $128 \times 128$ grid, so the learned law describes an ensemble of solution fields.
>
> This distinction is important because our approach can predict field-level ensemble statistics such as enstrophy (Figure 3b) and coherent-vortex structure (Figure 3a) and other observables that depend on the joint spatial organization across all grid points within one realization, which is unavailable to particle-level model in [1,2].
>
> Furthermore, we note that SCVM [2] assumes the governing PDE is known, whereas we learn population dynamics from data alone.
>
> ***[...] From a practical perspective, it is hard to place the proposed approach within the literature in terms of performance. [...] How do HOAM and DICE perform on the barotropic flow experiment?***
>
> Our comparisons are organized by regime:
> - In the high-dimensional setting, where the goal is to learn the population dynamics of full solution fields, the most relevant baseline is standard operator learning trained with an MSE loss, since this is the conventional machine-learning approach in that regime (see Figure 3). We also now include an additional experiment on a forced turbulence problem (JAX-CFD), which further strengthens this comparison (see response to reviewer TCfz).
> - In the lower-dimensional setting, we compare against AM, HOAM, and DICE because the evidence currently available in their own papers remains in this substantially lower-dimensional regimes (which agrees with our own empirical experiments). However, given the reviewer’s comment, in addition, we now additionally compare with JKONet* [3], both on an analytic potential example (see response to reviewer TCfz) and on the Vlasov example; these new results in the table below further clarify the relative strength of our approach compared to SOTA methods.
>
> Method  | two-stream || bump-on-tail ||
>  -- | -- | -- | -- | -- |
>  || avg. (↓) | final (↓) | avg. (↓) | final (↓)
> DICE | 4.5e-04 | 7.8e-04 | 6.5e-03 | 1.6e-02
> HOAM | 2.8e-03 | 4.0e-03 | 3.8e+01 | 5.5e+01
> AM  |4.6e+00 | 1.1e+00 | 2.5e-03 | 4.5e-03
> TPF (ours) | **3.2e-04** | **4.0e-04** | **3.1e-04** | **4.7e-04**
> JKONet* | 7.7e-03 | 9.8e-03 | 4.3e-3|	7.5e-3
>
> ***I am a bit puzzled about the comparison with the physics model in terms of kinetic energy (Figure 3b). At first glance I thought this constituted a violation of the underlying physics.***
>
> The quantity there is not the physical fluid kinetic energy contained in a single state, but the kinetic energy of the learned sample paths in state space ($||x_{t+1}-x_t||$). With that distinction in mind, Figure 3(b) shows exactly the effect we seek in this example: the learned population dynamics remove the unnecessary advective transport and vortex rotation that dominate the physics sample trajectories, while still preserving the slower population-level self-organization into fewer, larger coherent vortices. (See also answer to reviewer TCfz where we report additional physics quantities of interests that are well matched.)
>
> ***Does the method require the same number $N$ of observations for each time step?***
>
> No. Each time marginal is trained from its own independent sample set, so the sample count can vary by time step without changing the method.
>
> [3] A. Terpin et al., Learning diffusion at lightspeed, Neurips 2024

---

> > ### Author Rebuttal · Reviewer_zGm8 · 2026-04-03
> >
> > I would like to thank the authors for their detailed answers.
> >
> > My doubt concerning conditional normalizing flow stemmed from the fact that it sounded to me as a simpler way to model the problem (also within a single model). I understand now your point about scalability but I would be curious if in the barotropic example on the grid could be treated via interpolation as a continuos problem and then discretized again, enabling formal treatment with conditional normalizing fows (but this goes beyond the scope of your paper). I would like to thank the authors also for providing further experiments, which helped me contextualize the performance with other competing models.
> >
> > As most of my concerns have been addressed, I have raised my score to Weak accept.

---

> > > ### Author Response · Authors · 2026-04-06
> > >
> > > We thank the reviewer for the insightful comments and for the time to engage carefully with our response and additional experiments. We agree that further exploring the questions when to discretize is interesting. We also thank the reviewer that they will raise their score.

---

### Official Review · Reviewer_vbEq · 2026-03-09

**Soundness:** 3
**Presentation:** 2
**Significance:** 2
**Originality:** 3
**Overall Recommendation:** 4
**Confidence:** 2

**Summary:**

The paper considers a problem with independent snapshots of data at various times t. The goal is to learn a longitudinal ODE model which can produce trajectories matching these marginal samples.

The method is to train flow matching on all the snapshot data, amortised over all the t values. By sampling from this with a common initial value but various choices of t, data along a trajectory are produced. This can be used as training data to estimate the ODE gradient.

The paper contains supporting theory and examples on high dimensional data simulated from various models.

**Compliance With Llm Reviewing Policy:**

Affirmed.

**Final Justification:**

The follow-up comment makes a good case that the problem can't just be solved by rerunning trajectory simulators. It also describes some abstract problems which the method can solve. But I'm still unclear what the eventual practical applications would be. On balance the rebuttal and follow-up comment have addressed my concerns enough for me to raise my score to 4 "weak accept".

**Key Questions For Authors:**

Can you address the numbered points in the "Main weaknesses" section? The ones on significance and presentation would be most useful to increase my score to 4.

**Limitations:**

Yes

**Strengths And Weaknesses:**

Below I use bold for comments on the 4 criteria mentioned in the review form.

## Strengths

The proposed solution is simple, powerful and seems novel to me (**originality**). The paper provides nice theoretical guarantees for it (**soundness**), and the examples show good behaviour on high dimensional problems.

## Weaknesses

### Main weaknesses

1. When do the conditions for the theory hold?
  - Does the $v$ defined in Proposition 2.4 meet the conditions of the other results?
  - Does $v$ produced from the method in Section 3 meet the conditions of the results?

2. Can you say more about the practical motivation for the method? (**significance**) The example sections had good case studies of developing fast simulators with good behaviour. However they seem slightly artificial because the underlying models could be used to simulate trajectory data for training, not just snapshot data.

3. If I understand correctly, then OT methods are only used on the simple Gaussian mixture example. Is it feasible to implement them for the other examples to demonstrate that the proposed method is superior?

4. Parts of the paper are hard to follow - see "explanations" section below (**presentation**).

### Minor weaknesses (explanations)

* Line 25 (right column): "making a... regression... inherently rough". What does this mean?
* Figure 1 is on page 2, but I think the notation it uses isn't introduced until Section 2.1.
* The terms "physics-time" and "sampling-time" are used throughout the introduction, but only defined on page 4.
* I initially found Figure 2 hard to understand. Is the idea that there are 3 columns for $t_1<t_2<t_3$? If so maybe you could indicate this in the figure and caption.
* Line 162 (left column): I guess a non-gradient flow would be one that can't be represented as a gradient of some function. But can you expand on what "non-gradient components" are?
* Line 262 (left column): What's a $H^1$ norm?
* Line 221 (right column): What's $I_s(t)$ and how is it related to $I_s(a,x_t)$?
* Figure 3: The TPF output (bottom row) looks quite different to the physics model output (top row). Is this a problem?
* Line 425 (left column): "We see that the HOAM... and DICE... methods... do not manage to capture the fine structures of the phase-space filamentation". Can you explain what differences to look for? I can see the HOAM and DICE images are slightly more blurry for the "two-stream" example. But I can't see much difference for the "bump-on-tail" example.

### Minor weaknesses (miscellaneous)

* Page 1. Are some conditions on $\rho(t)$ needed for an admissible velocity field to exist? For example what if $\rho(0)$ and $\rho(t)$ for $t>0$ have different supports?

---

> ### Author Rebuttal · Authors · 2026-03-30
>
> We thank the reviewer for a careful and detailed reading of the paper.
>
> ***"Do the $v$s [...] meet the conditions of the results?"***
>
> Yes, the $v$ defined in Prop. 2.4 meets the assumptions as long as the target marginal curve $t \mapsto \rho(t)$ is regular. A regular $v$ leads to a regular flow $\Phi$, this is shown in Prop. 2.1.
> Using the results from [1], we can express $\rho_I(x,t,s)$, the density of the stochastic interpolant, as a function of $\rho(x, t) \propto \exp (-V(x,t))$:
> $$\rho_I(x,t,s) = \log \int \exp (-V(y,t) - \frac 1 2 \frac{\beta_s^2}{\alpha_s^2} |y|^2 + \frac{\beta_s}{\alpha_s^2} y \cdot x ) dy$$
> The connection between $v$ and $\nabla \log \rho_I$ is stated in Prop. 2.4. In a revision, we will add this argument to make clear that the regularity of $v$ depends entirely on that of $\rho$, with approximation error controlled by the expressivity of the architecture and the flow matching training loss.
>
> ***"Can you say more about the practical motivation? / The TPF output looks quite different to the physics model"***
>
> Even when trajectory data are available, learning to reproduce individual paths can be both difficult and unnecessary when downstream tasks depend only on statistical quantities. In complex systems such as chaotic or turbulent dynamics, trajectories are often too irregular to be reliable learning targets, whereas the corresponding population dynamics are typically much smoother and easier to model. **The goal of the method is not to reproduce the individual sample trajectories.**
>
> We compare our method to a neural operator that learns sample trajectories. Using the same computational budget, our method outperforms direct learning substantially.
> Our approach builds population-level surrogates for applications driven by ensemble statistics, such as control and design under uncertainty or parameter inference, while also enabling repeated ensemble predictions at far lower cost than full physics simulations, with speedups exceeding 100x in the barotopic-flow example.
>
> ***"Is it feasible to implement [OT methods] for the other examples?"***
>
> The feasibility depends on the spatial dimension as well as the number of samples.
> Action Matching and related methods struggle in large spatial dimensions. This is evident from the rather limited $32 \times 32$ resolution they operate on (DICE, AM) and is also discussed in Appendix C of Neklyudov et al.
> Computing a piece-wise OT coupling as in Terpin et al. (JKONet*, [2]) requires $\mathcal{O}(n^3 T)$ resources where $n$ is the number of samples and $T$ the number of time-steps.
>
> Given the reviewer’s comment, we added new experiments including an additional baseline, JKONet*, and a new gradient-flow example (Blickhan et al., 2025; Section 8.2). The first table shows that our method performs strongly on gradient flows, outperforming the OT-based JKONet* and DICE. The second table demonstrates that on the Vlasov example, our TPF approach again achieves higher accuracy than JKONet*. This is consistent with known limitations of OT, such as their restriction to gradient dynamics, sensitivity to cost design, and the computational difficulty of finding optimal couplings.
>
> **W2 distances (gradient-flow example)**
> |Time|TPF (ours)|JKONet*|DICE|
>  --|-- | -- | -- |
> |0.00|0.074|0.074|0.128|
> |0.05| 0.111|0.101|0.127|
> |0.20| 0.119|0.133|0.136|
> |0.50| 0.150|0.459|0.156|
> |1.00| 0.122|0.545|0.234|
>
> **W2 distances (Vlasov example)**
> Method | TS avg.↓|TS final↓|BOT avg.↓|BOT final↓
> --|--|--|--|--
> DICE |4.5e-4|7.8e-4|6.5e-3|1.6e-2
> HOAM |2.8e-3|4.0e-3|3.8e+1|5.5e+1
> AM|4.6e+0|1.1e+0|2.5e-3|4.5e-3
> TPF  |**3.2e-4**|**4.0e-4**|**3.1e-4**|**4.7e-4**
> JKONet*|7.7e-3|9.8e-3|4.3e-3|7.5e-3
>
> We will address all the “minor weaknesses” in a revision but provide here answers only to a selected list due to space constraints:
> - Visual difference in Figure 3: the TPF output matches the correct marginal distribution, not individual trajectories -- we will clarify this in the caption.
> - "inherently rough" refers to the regression target being a finite-difference approximation of a non-differentiable path -- we will rewrite this sentence for clarity.
> - Figure 3: We agree that the differences are not as easy to spot, but Table 1 does show the improved performance of our method.
> - Non-gradient components: Any vector field $u$ on a simply connected domain with suitable boundary conditions can be de-composed as $\nabla \phi + \omega$ where these two components are $L^2(\rho)$-orthogonal. This implies that $\nabla \cdot (\rho \omega) = 0$. We refer to $\omega$ as the "non-gradient component" of $u$.
> - $I_s(t)$ is the interpolant at sampling time $s$: $I_s(t): (a, x_t) \mapsto I_s(a, x_t)$. So $I_s(a, x_t)$ is $I_s(t)$ evaluated at the noise/data pair $(a, x_t)$.
>
> [1] Daniels: On the Contractivity of Stochastic Interpolation Flow, arXiv:2504.10653
>
> [2] A. Terpin et al., Learning diffusion at lightspeed, Neurips 2024

---

> > ### Author Rebuttal · Reviewer_vbEq · 2026-04-02
> >
> > Thanks for the helpful replies and extra simulation example.
> >
> > ## Q1
> >
> > The reply about Proposition 2.4 resolves that comment. What do you think about the question regarding Section 3?
> >
> > ## Q2
> >
> > I'm still unclear about the practical motivation. I think there's likely some background context about the applications you have in mind which I'm not familiar with.
> >
> > *"Even when trajectory data are available, learning to reproduce individual paths can be both difficult and unnecessary..."*
> >
> > In the examples it seems you already have fast simulators, so surely you could just sample individual paths as needed, rather than needing to learn to reproduce them.
> >
> > *"...when downstream tasks depend only on statistical quantities."*
> >
> > Could you elaborate on what kind of practical downstream tasks you have in mind?
> >
> > ## Q3,Q4,minor comments
> >
> > The responses address all my comments here.

---

> > > ### Author Response · Authors · 2026-04-03
> > >
> > > *The reply about Proposition 2.4 resolves that comment. What do you think about the question regarding Section 3?*
> > >
> > > Thank you for the follow-up. Regarding Section 3, the answer is yes: For the method in Section 3, $v: (x, t, s) \mapsto v(x,t,s)$ is parametrized by a neural network. Using smooth activation functions (e.g. swish), this is a $C^\infty$ function by design. Through the training, $v_\theta(x, t, s)$ will be a smooth approximation to the velocity field $v$ from Proposition 2.4.
> > >
> > > *I'm still unclear about the practical motivation. I think there's likely some background context about the applications you have in mind which I'm not familiar with. [...] In the examples it seems you already have fast simulators, so surely you could just sample individual paths as needed, rather than needing to learn to reproduce them.*
> > >
> > > We appreciate the reviewer’s question and would like to clarify the practical regime we target. In our experiments, **we do not have access to a fast simulator**, we need to solve the PDEs numerically, which is expensive. For example, in the barotropic-flow experiment, the PDE solver must resolve the true dynamics using small time steps due to the CFL condition and high spatial resolution, which leads to a significant computational cost for each new simulation.
> > >
> > > TPF is useful precisely because it learns a **fast surrogate model for the evolution of the distribution**, enabling us to replace repeated ensemble PDE simulations when the quantities of interest are statistical rather than pathwise (we show 100x speedups in Figure 3d).
> > >
> > > This is also why learning a classic surrogate model of the direct trajectories (paths) is not the right objective in the settings we study. In chaotic or turbulent systems, accurate long-time prediction of individual trajectories/paths with fast motion is both substantially harder and often unnecessary: trajectory-based surrogates (such as operator learning, and MSE-fitted models as in Figure 3) must spend capacity reproducing fast, realization-specific motion that is not actually relevant for the quantities of interest. In contrast, many downstream tasks depend on distributional statistics such as enstrophy (Figure 3). This is exactly what we see in Figure 3: the MSE-fitted trajectory model is the natural trajectory-based surrogate baseline (“operator learning”), but it does not reproduce the relevant statistical behavior of enstrophy nearly as well as TPF.
> > >
> > > *Could you elaborate on what kind of practical downstream tasks you have in mind?*
> > >
> > > We will expand the discussion in the paper to emphasize that the regime we target is **many-query settings** where one must repeatedly propagate ensembles over time, often for new initial conditions or new parameter values. The paper states this goal in the introduction and in the inference section but we will make this clearer: learn dynamics that can be used for prediction at unseen physics parameters, and then integrate only the learned population dynamics at test time, with low costs (one network evaluation per time step).
> > >
> > > Examples of such many query settings include: choosing design or control parameters under uncertainty using expected performance, risk, or probabilities of undesirable events; inverse problems / parameter identification by matching observed marginal statistics or summary observables; and repeated parameter sweeps, optimization loops, where the same expensive PDE solver would otherwise have to be run for a large ensemble many times, which is intractable. This is also consistent with the paper’s parametric setting: the learned model is designed to generate ensembles for new initial conditions and parameters, and in the Vlasov-Poisson example it is evaluated on unseen Debye-length parameters.

---

### Official Review · Reviewer_TCfz · 2026-03-13

**Soundness:** 4
**Presentation:** 4
**Significance:** 3
**Originality:** 3
**Overall Recommendation:** 4
**Confidence:** 4

**Summary:**

The paper address the problem of learning population dynamics of physics based dynamical system from time-marginal samples using two stages -- conditional flow matching and then learning a physics-time velocity. The theory is based on optimal transport and two-parameter flows. The paper establishes a rigorous framework to support their claims that are validated by very strong benchmark tests that demonstrate high state space result.

**Compliance With Llm Reviewing Policy:**

Affirmed.

**Final Justification:**

The rebuttal addressed my main concerns and the authors have an extremely creative way of formulating the problem linking conditional flow mapping to dynamics inference. My position is still that it should be accepted and my assessment remains unchanged.

**Key Questions For Authors:**

### Questions

1. Can the authors include one setting where the underlying dynamics is much closer to gradient dynamics, this will clarify the two-parameter flow setting is not only tied to rotational/non-gradient type dynamics.

2. The barotropic flow is good, but it doesn't itself probe whether the learned physics-time velocities are meaningful physically. Can the
framework be extended for something like a lid-driven cavity flow, or flow past a cylinder example?

**Limitations:**

Yes.

**Strengths And Weaknesses:**

### Strengths

1. The paper is extremely well written and provides a unique viewpoint to link conditional flow mapping to dynamics inference. This is an interesting conceptual link that
charactirizes inference in chaotic/stochastic systems.

2. Compared to most papers with similar goals, this paper is rigorously grounded and proves that given a class of two-parameter transport representation the learned (regressed) velocity field is  uniquely determined
from the marginals.

3. The benchmark experiments are meaningful and show that the often used (scalar potential/gradient) assumptions can be restrictive. Moreover, the barotropic example shows scaling to largeish state space dimensions.

### Weaknesses

1. AI novelty is lacking. The novelty comes in the framing of the problem which is very interesting, but the use of AI is very standard. One potential way this could have been strengthened is incorporating structure in the
learned physics-time velocity field (divergence free, Poisson type etc.) which leads to the next related point.

2. The learnt physics-time velocity field is uniquely determined by the marginals under the two-parameter representation. It is not constrained to satisfy the underlying physics, so it is not clear how robust is the learnt
velocity to the choices of these representation.

3. Benchmarks show the method winning in settings where there is non-gradient structure. This is good as it is the main claim of the paper, but this is where structured methods can outperform gradient based OT method. Is the proposed structure better outside this regime as well?

4. While the theory is solid, I am not entirely sure if the assumptions truly match the paper's motivated regime: chatoic/stochastic systems. For example, the theory is still $C^2$ diffeomorphic (a globably smooth flow map from the base to marginals at all timess). What happens when we have sharp structures, multiple attractors etc?

---

> ### Author Rebuttal · Authors · 2026-03-30
>
> > The novelty comes in the framing of the problem which is very interesting, but the use of AI is very standard.
>
> We agree that the novelty comes through the learning formulation: we show how a conditional generative transport model can be converted into a model for physical-time dynamics, which conditional flow matching by itself does not provide. Moreover, we prove that the extracted velocity is **unique, satisfies the continuity equation, and inherits regularity** from the learned base-to-marginal transport. Overall, our approach **expands the learnable class beyond gradient/minimal-energy dynamics** and **makes regimes practical where OT/JKO-style approaches are restrictive or computationally prohibitive**, including our high-dimensional barotropic example and the **newly added forced turbulence example**.
>
> > [...] non-gradient structure... Is the proposed structure better outside this regime as well?
>
> We ran the method on the gradient-flow example in Blickhan et al, 2025 (Section 8.2, arXiv 2507.05107), where the underlying dynamics are gradient. Our method outperforms DICE as well as JKONet* [1] especially at later times t > 0.20, confirming that our two-parameter flow is well suited even when gradient structure is present.
>
> **W1 distance**
> | Time | TPF (ours) | JKONet* | DICE |
> |-----:|:------:|:------:|:------:|
> | 0.00 | 0.006 | 0.006 | 0.105 |
> | 0.05 | 0.072 | 0.061 | 0.091 |
> | 0.20 | 0.083 | 0.095 | 0.111 |
> | 0.50 | 0.074 | 0.286 | 0.156 |
> | 1.00 | 0.036 | 0.353 | 0.047 |
>
>
> **W2 distance**
> | Time | TPF (ours) | JKONet* | DICE |
> |-----:|:------:|:------:|:------:|
> | 0.00 | 0.074 | 0.074 | 0.128 |
> | 0.05 | 0.111 | 0.101 | 0.127 |
> | 0.20 | 0.119 | 0.133 | 0.136 |
> | 0.50 | 0.150 | 0.459 | 0.156 |
> | 1.00 | 0.122 | 0.545 | 0.234 |
>
> > The learned physics-time velocity field is [...] not constrained to satisfy the underlying physics [...] The barotropic flow is good, but it doesn't itself probe whether the learned physics-time velocities are meaningful physically. [...]
>
> We thank the reviewer for this important comment. Our goal is not to recover the unknown underlying physics velocity of the trajectories but a velocity that describes the population dynamics of the solutions of the physics problem. This is exactly the notion of correctness studied in the paper: we prove that, once a sufficiently regular two-parameter flow is fixed, the learned velocity field is well defined and unique and solves the continuity equation. Therefore, the **learned dynamics are proveably consistent with the observed marginal evolution of the underlying physics problems.**
>
> In this setting, the most relevant notion of “physically meaningful” is whether the **samples** generated by the learned model are physically meaningful as an ensemble, rather than interpreting the learned velocity field. Accordingly, we **assess physical fidelity through quantities of interest that depend on the sample distribution.** If these QoIs are accurately approximated by the generated ensemble, then the samples are physically meaningful for downstream tasks (e.g., optimization/design, inverse problems). In the barotropic setting, we report, for example, enstrophy that characterizes the evolving sample ensembles, which is approximated well.
>
> To further emphasize this point, we applied our approach to a **new example: Kolmogorov flow** (details under “forced turbulence” in [2] and JAX-CFD on github). We discretize on 128x128 and learn a two-parameter flow model analogous to our barotropic example. In this example, the energy spectrum should decay as $k^{-3}$, where $k$ is the frequency (see [2]). We obtain the following exponents $m$ for $k^{-m}$ when fitting to the spectrum decay of the generated samples (averaged over 16 runs)
>
> | Timestep | Physics model | Operator learning (MSE-fitted) | TPF (ours) |
> |--|--|--|--|
> | 32 | -2.8008 | -4.1643 | -2.7127 |
> | 64 | -2.8797 | -3.9289 | -2.8941 |
> | 95 | -2.9320 | -3.8183 | -2.8904 |
> | 127 | -2.9709 | -3.8058 | -2.8808 |
>
> Our TPF matches the exponent -3 closely, which again shows that the generated samples match the physics property expected in this case (i.e., physically meaningful). One can also see that the operator-learning-based model (MSE-fitted) has a much harder time getting the energy spectrum decay right because trying to capture the trajectories of this chaotic/turbulent flow is difficult.
>
> > What happens with sharp structures, multiple attractors, etc.?
>
> The $C^2$ diffeomorphism assumption is standard and is needed for the theoretical guarantees  (see, e.g., Albergo et al., arXiv 2303.08797 on theory for flow models, Definition 1 and Assumption 5). In practice, modern flow architectures approximate sharp structures well at finite resolution, and our experiments include examples with non-trivial attractor geometry.
>
> [1] Terpin et al., Learning diffusion at lightspeed, Neurips 2024
>
> [2] Kochkov et al. Machine learning–accelerated computational fluid dynamics, PNAS 2021

---

> > ### Author Rebuttal · Reviewer_TCfz · 2026-04-03
> >
> > Thank you! My questions have been answered. This is a good contribution to the field and the rebuttal answers my skepticisms. However to be broadly applicable to the physics, engineering and science community an extension of the method to get physics relevant population velocity would make this contribution unmistakable even though I understand the point the authors are raising.

---

> > > ### Author Response · Authors · 2026-04-06
> > >
> > > Thank you for this very encouraging assessment and for highlighting this direction about physics-informed velocity fields. We fully agree that an important next step is to enrich our approach with additional structure so that the selected population velocity also reflects domain-relevant physics principles, such as conservation laws, symmetries, and application-specific constraints.
> > >
> > > We believe that with this current work we contribute the scalable backbone that gives a principled and computationally practical starting point on which one can then impose further physics-informed selection criteria. In particular, Proposition 2.2 (and Section 2.2) establish an explicit correspondence $v \mapsto \Phi \mapsto u$, which means that constraints on the velocity field $u$ can be systematically translated backwards into corresponding constraints on $\Phi$, and hence on the base-to-marginal transport $v$. This makes physics-informed extensions a natural and technically tractable next step within our approach.
> > >
> > > As an example, consider the following physics-informed constraint: We want to enforce a divergence-free field $\nabla \cdot u(x, t, s=1) = 0 \; \forall t, x$.
> > >
> > > This can be enforced by constraining the $v$s: $\nabla \cdot v(x, t, s) = 0 \, \forall x, t, s$ and $\nabla \cdot u(x, t, s=0) = 0 \; \forall x, t$ implies $\nabla \cdot u(x, t, s=1) = 0 \; \forall x, t$. Note that $u(x, t, s=0) \equiv 0$ in our method, so we only need to enforce $\nabla \cdot v \equiv 0$. We prove this below.
> > >
> > > We also would like to mention that a lot of existing literature considers gradient-form $u$s as they are inspired by optimal transport theory (for example, JKONet*). This minimal-kinetic-energy selection rules out other, physics-informed, selection criteria.
> > >
> > > We appreciate the emphasis on the physics-informed velocity fields, and we will make this future work direction more explicit in a revision.
> > > ___
> > > Proof that $\nabla \cdot v(x, t, s) = 0 \, \forall x, t, s$ implies $\nabla \cdot u(x, t, s=1) = 0$:
> > > Take the divergence of the compatibility condition
> > > $$ \nabla \cdot ( \partial_s u + v \cdot \nabla u - u \cdot \nabla v - \partial_t v ) = 0 .$$
> > > Assuming everything is regular enough to commute derivatives, this implies
> > > $$ \partial_s \nabla \cdot u + v \cdot \nabla (\nabla \cdot u) - u \cdot \nabla (\nabla \cdot v) - \partial_t (\nabla \cdot v) = 0 .$$
> > > The last two terms vanish when $\nabla \cdot v \equiv 0$. Hence,
> > > $$ ( \partial_s + v \cdot \nabla )(\nabla \cdot u) = 0. $$
> > > But this implies that $\frac{d}{ds} (\nabla \cdot u)(\Phi_{t,s}(a)) = 0$ for all $\Phi_{t,s}(a) = x_{t,s}$, hence $\nabla \cdot u = 0$ is preserved along the flow. Hence,
> > > $$ (\nabla \cdot u)(x, t, s=1) = (\nabla \cdot u)(\Phi_{t,s=1}^{-1}(x), t, s=0) = 0 \quad \forall x, t $$
> > > As claimed.

---

### Official Review · Reviewer_ChK2 · 2026-03-13

**Soundness:** 3
**Presentation:** 2
**Significance:** 2
**Originality:** 3
**Overall Recommendation:** 4
**Confidence:** 3

**Summary:**

The paper studies the problem of learning dynamical systems from population-level data, where only unlabeled samples from the distribution of the system state at different times are available. The proposed approach introduces a generative model based on two-parameter flows. The model maps a latent base distribution to the observed marginal distribution at each time using a conditional flow matching framework, and then extracts a corresponding velocity field that represents the underlying physical-time dynamics.

The key idea is to couple all the marginal distributions through the latent distribution, that is to say, the same initial sample from latent distribution is used to generate sample from the marginal distributions. In this way, a least-squares like criteria can be used to learn the dynamics.

The resulting model provides a way to generate trajectories consistent with the observed population. The method is demonstrated on several examples, including high-dimensional dynamical systems

**Compliance With Llm Reviewing Policy:**

Affirmed.

**Final Justification:**

I raised my score after reading the rebuttal and addressing the significance concerns and adding additional numerical experiments. The paper is also proposing an original method.  However,  my concerns about the presentation remain.

**Key Questions For Authors:**

None.

**Limitations:**

It is not substantial.

**Strengths And Weaknesses:**

__Soundness__:  Overall, the paper appears to be technically sound. The methodological framework is clearly structured and the theoretical arguments seem consistent with the proposed modeling approach. To the extent that I checked, the technical derivations and claims appear reasonable. However, many of the more detailed arguments and proofs are presented in the appendix, and I did not verify all of them carefully.

__Presentation__: Overall, the presentation of the paper is acceptable and the main ideas can be followed without too much difficulty. That said, there are several aspects of the presentation that could be improved to make the motivation and narrative clearer.
- While it is acceptable for the introduction to contain technical material, this should ideally be reserved for concepts that are essential for motivating the work. The random walk example presented early in the introduction is theoretically interesting, but it does not clearly illustrate the practical motivation for learning population dynamics or the limitations of learning from individual sample trajectories. In particular, the example emphasizes the non-differentiability of sample paths, but in practice one can still fit stochastic models to sample trajectories and estimate parameters (such as diffusion coefficients) through optimization. My understanding is that the motivation for population dynamics learning is not necessarily that trajectory-based learning is fundamentally flawed, but rather that in many applications trajectory-level data is not available.
- The practical implications of learning the dynamics are not sufficiently discussed. It would be helpful to explain more concretely in what types of applications one would want to learn such dynamics and how the learned model can be used afterward.
- Finally, the organization of the paper could be improved. Providing a brief outline of the paper near the end of the introduction would help guide the reader. In addition, the flow between sections could be made smoother. Currently, the paper contains many subsections that sometimes feel somewhat isolated, and their titles do not always clearly indicate their role in the overall development.

__Significance__:
- At present, the significance of the paper is not clearly demonstrated. The problem of learning dynamics from population-level data is potentially important. However, the paper does not sufficiently highlight why this problem is important in practice or how the proposed method advances the state of the art in a meaningful way. One possible way to strengthen this aspect would be to draw more explicitly on the existing literature on population dynamics learning and related problems.
- In addition, the paper would benefit from a more detailed comparison with one or two state-of-the-art approaches addressing similar problems. This could include either theoretical discussion or empirical comparisons that clarify the advantages and limitations of the proposed method relative to existing techniques. Some of this is already discussed in the numerical experiments, but it would be good to be highghted or expanded

__originality__: The key idea underlying the proposed methodology—coupling the time marginals through a shared latent distribution—is interesting and, to the best of my knowledge, appears to be original. Rather than directly estimating couplings between observed distributions at different times, the method introduces a latent generative mechanism that maps a base distribution to each marginal distribution. This provides a structured way to relate the marginals and infer the underlying dynamics without requiring labeled trajectories or explicit pairings between samples across time.

---

> ### Author Rebuttal · Authors · 2026-03-30
>
> > [...] how the proposed method advances the state of the art in a meaningful way
>
> The main point is that our contribution is not only a new way to learn population dynamics, but a **method that reaches a qualitatively different application regime**: solution fields of physical systems with fast inference and non-gradient dynamics.
>
> We achieve this by first substantially extending the **scale** at which population dynamics can be learned. In the papers most closely related to ours, the largest reported examples we found are $d=50$ for JKOnet* [1], $d=100$ for HOAM, $d=1024$ for DICE, and $d=3072$ for AM. By contrast, our barotropic-flow experiment operates on a $128 \times 128$ vorticity grid, i.e. $d=16384$. This is the regime relevant for turbulent and other flow systems. Our method reaches this regime by leveraging a scalable conditional-flow-matching backbone for the base-to-marginal transport, and then extracting a single explicit physics-time velocity for enabling fast downstream inference.
>
> Second, our method is **not tied to gradient-flow / minimal-energy dynamics**. JKOnet*, DICE, HOAM, and AM all learn gradient flows or optimal-transport-based dynamics. In contrast, our approach learns an admissible physics-time velocity induced by the two-parameter flow and therefore does not force the dynamics to be gradient. This is important in applications with intrinsic rotational or circulating behavior, such as vortex-dominated fluid systems as the barotropic example, where restricting the dynamics to a gradient field can be too limiting.
>
> > Missing motivation for learning population dynamics
>
> We agree with the reviewer that we should have made the importance of learning population dynamics clearer in the paper. The main motivation is that many chaotic and turbulent physical systems have **individual trajectories that are simply too irregular and complex to serve as a robust target for learning** (see the MSE-fitted operator learning results in Figure 3). By contrast, the associated **population dynamics are often much smoother** and therefore more amenable to learning.
>
> The **learned model serves as a surrogate for the time-evolution of an ensemble of solutions**, rather than for individual trajectories. This **population-level surrogate is useful in downstream tasks precisely because many decisions are made from ensemble statistics, not individual trajectories**. For example, in optimal design or control under uncertainty, one often wants to choose parameters or controls that optimize an objective depending on expected performance, risk, or probability of desirable events. In inverse problems / parameter inference, one often matches summary statistics to model predictions in order to identify unknown parameters. Our population-dynamics models enable repeated ensemble predictions at much lower cost than full physics simulations (e.g., speedups of $>100\times$ in the barotropic flow example).
>
> > A more detailed comparison with one or two state-of-the-art approaches would strengthen the paper
>
> We note that we already compare to HOAM, DICE, and AM as well operator learning (MSE-fitted trajectory model). Following the reviewer’s comment, we added three new experiments:
> - We now compare to JKOnet* [1] on the Vlasov example. In the table below we report the same results as in the paper and additional JKOnet*, which provides further evidence that our TPF approach performs well
>
> Method  | two-stream || bump-on-tail ||
>  -- | -- | -- | -- | -- |
>  || avg. (↓) | final (↓) | avg. (↓) | final (↓)
> DICE | 4.5e-04 | 7.8e-04 | 6.5e-03 | 1.6e-02
> HOAM | 2.8e-03 | 4.0e-03 | 3.8e+01 | 5.5e+01
> AM  |4.6e+00 | 1.1e+00 | 2.5e-03 | 4.5e-03
> TPF (ours) | **3.2e-04** | **4.0e-04** | **3.1e-04** | **4.7e-04**
> JKONet* | 7.7e-03 | 9.8e-03 | 4.3e-03  | 7.5e-03
>
> - We compare on a new gradient flow example from [Blickhan et al., 2025 (Section 8.2)] and show that TPF is performing well even in this setting that is well suited for potential-based and OT-based techniques (see also reviewer TCfz). The table below reports W2 distance.
>
> | Time | TPF | JKONet* | DICE |
> |--|--|--|--|
> | 0.00 | 0.074 | 0.074 | 0.128 |
> | 0.05 | 0.111 | 0.101 | 0.127 |
> | 0.20 | 0.119 | 0.133 | 0.136 |
> | 0.50 | 0.150 | 0.459 | 0.156 |
> | 1.00 | 0.122 | 0.545 | 0.234 |
>
> - We added another turbulent flow experiment (see answer to TCfz for details), where our approach predicts well the exponent 3 in the $k^{-3}$ energy spectrum decay, while operator learning struggles here due to the chaotic/turbulent nature:
>
> | Timestep | Physics model | Operator learning (MSE-fitted) | TPF (ours) |
> |--|--|--|--|
> | 32 | -2.8008 | -4.1643 | -2.7127 |
> | 64 | -2.8797 | -3.9289 | -2.8941 |
> | 95 | -2.9320 | -3.8183 | -2.8904 |
> | 127 | -2.9709 | -3.8058 | -2.8808 |
>
>
> These new experiments provide further evidence that our TPF achieves SOTA accuracy while being scalable to regimes beyond the reach of current methods.
>
> [1] Terpin et al., Learning diffusion at lightspeed, Neurips 2024

---

> > ### Author Rebuttal · Reviewer_ChK2 · 2026-04-03
> >
> > Thanks for the response. My concerns to significance are almost addressed, but not about the presentation.

---

> > > ### Author Response · Authors · 2026-04-03
> > >
> > > Thank you for acknowledging our response. You are correct in that we mostly focused on your significance concerns in the rebuttal so far. However, your comments about the presentation are also very much appreciated. We will use the additional space in the camera-ready draft to improve readability.
> > >
> > > In particular, we will make the following updates to our article:
> > >
> > > 1. Clearly state the motivation for learning population dynamics. They are twofold:
> > >     - In some applications, trajectory-level data is not available and hence learning population-level dynamics is the only option.
> > >     - Even when trajectory data is available, it can be beneficial to match the data only on the population level, not the trajectory level. This can lead to simpler learning problems and a speed up in the inference step, as shown in Figure 3.
> > >
> > > 2. State the practical application of learning these surrogate dynamics emulators.
> > >     - We will clarify that our setting is many-query scenarios, where ensembles must be repeatedly propagated for new initial conditions or parameter values. The goal is to learn dynamics that generalize across parameters and can be integrated at test time at low cost (one network evaluation per time step).
> > >     - Typical use cases include design or control under uncertainty, inverse problems via matching marginal statistics, and repeated parameter sweeps or optimization loops where full PDE simulations would be prohibitively expensive. This aligns with our parametric setting, where the model generates ensembles for unseen conditions.
> > >     - In the Vlasov–Poisson equation example, we will clearly state this relation to generalization over unseen parameters (in this case, the Debye length).
> > >
> > >
> > > 3. Remove the random walk example from the introduction. We believe this example can be helpful to illustrate the second motivation to learn population dynamics, but we will move it to a less prominent position and importantly after the motivation for population dynamics inference has been stated.
> > >
> > >
> > > 4. Paper outline: We will provide an outline of the paper that includes pointers to our main results at the end of the introduction.
> > >
> > > 5. Flow: We will consolidate some of the subsections to improve the flow of the paper especially in Section 2. At the beginning of every section, we will add a short summary of what the main points of presentation will be in this section.

---

### Decision · Program_Chairs · 2026-04-30

**Decision:**

Accept (regular)

**Comment:**

The consensus among reviewers is that the paper presents a technically sound and original methodology for learning population dynamics from marginal snapshots. While reviewers initially raised concerns regarding the clarity and the thoroughness of empirical validation, these concerns was mostly addressed by the authors' rebuttal with e.g. new experiments against new baseline methods like JKONet* and clearer application use cases. Overall due to the reviewer consensus and no major concerns I recommend accept.